# Quadrotor Formation Strategies Based on Distributed Consensus and Model Predictive Controls

**Chia-Wei Chang and Jaw-Kuen Shiau ***

Department of Aerospace Engineering, Tamkang University, Tamsui, New Taipei City 25137, Taiwan; chiaweiichang@gmail.com
**\*** Correspondence: shiauj@mail.tku.edu.tw

**Abstract:** In this study, the distributed consensus control and model predictive control (MPC)-based formation strategies for quadrotors are proposed. First, the formation-control problem is decoupled into horizontal and vertical motions. The distributed consensus control and MPC-based formation strategy are implemented in the follower's horizontal formation control. In the horizontal motion, the leader tracks the given waypoints by simply using the MPC, and generates the desired formation trajectory for each follower based on its flight information, predicted trajectory, and the given formation pattern. On the other hand, the followers carry out the formation flight based on the proposed horizontal formation strategy and the desired formation trajectories generated by the leader. In the vertical motion, formation control is carried out using only the MPC for both the leader and the follower. Likewise, the leader tracks the desired altitude/climb rate and generates the desired formation trajectories for the followers, and the followers track the desired formation trajectories generated by the leader using the MPC. The optimization problem considered in the MPC differs for the horizontal and vertical motions. The problem is formulated as a quadratic programming (QP) problem for the horizontal motion, and as a linear quadratic tracker (LQT) for the vertical motion. Simulation of a comprehensive maneuver was carried out under a Matlab/Simulink environment to examine the performance of the proposed formation strategies.

**Keywords:** quadrotor formation control; consensus control; model predictive control; linear quadratic tracker; quadratic programming

---

## 1. Introduction

As the quadrotor becomes increasingly popular and easy to acquire, it has gained much attention from researchers and manufacturers. Along with the various well-developed control algorithms of a single quadrotor, researchers have shifted their attention toward formation-control algorithms in recent years. A survey [1] classifies the formation flying control algorithms of spacecraft into five architectures: multiple-input and multiple-output, leader–follower, virtual structure, cyclic, and behavioral. The above classifications can be extended to the formation-control algorithms of the quadrotor. Among the above formation-control architectures, the leader–follower architecture [2–6] has received most attention from researchers.

In this study, the distributed consensus control and model predictive control (MPC)-based formation strategies considering the leader–follower architecture are proposed. First of all, the formation-control problem is decoupled into horizontal and vertical motions. In the horizontal motion, the leader tracks the given waypoints by using the MPC, and generates the desired formation trajectory for each follower based on its flight information, predicted trajectory, and the given formation pattern. The followers carry out the formation flight based on the proposed horizontal formation strategy and the desired formation trajectories generated by the leader. The follower's horizontal

formation strategy is set up based on the distributed consensus control and the MPC. Moreover, the optimized problem, which is considered in the MPC as adopted by the leader's tracking control and the follower's formation strategy in the horizontal motion, is formulated as a quadratic programming (QP) problem. In the vertical motion, the leader tracks the desired altitude and climb rate and generates the followers' desired formation trajectories and the followers track the desired formation trajectories generated by the leader using the MPC. The optimized problem, which is considered in the MPC as adopted by the leader and the follower in the vertical motion, is formulated as a linear quadratic tracker (LQT).

Section 2 provides the background of this study. Both the linear and nonlinear equations of motion (EOMs) of the quadrotor are derived in Section 3. Section 4 briefly introduces the control laws that are adopted in the formation strategies. The full map of the formation strategies for the leader and the follower are carried out in Section 5. In Section 6, the preparation for the simulation, including the gains for the control laws as well as the Matlab/Simulink environment setup (Matlab R2016b, The MathWorks, Inc., Natick, MA 01760, USA), is provided. A simulation of the comprehensive maneuver was carried out to examine the performance of the formation strategies, and simulation results are illustrated in Section 7. Section 8 discusses the simulation results and the performance of the proposed formation strategies. The conclusions of the study are summarized in Section 9.

## 2. Background

Various methodologies of formation control have been proposed in recent years. For example, the backstepping method considering the nonlinear quadrotor model [7,8], linear quadratic control considering the linear quadrotor model [2], consensus control, and MPC [9]. In the field of consensus formation control, a comparative review of the latest trends has been carried out in Reference [10]. The author of Reference [11] provides a theoretical framework for analyzing the consensus algorithms for multiagent networked systems. Furthermore, a consensus algorithm for a second-order system is proposed in Reference [12].

This study is motivated by the results in Reference [13], in which the authors propose a formation-control method using decentralized MPC and consensus-based control. First, the leader computes all of the followers' reference trajectories using a consensus-based algorithm, and transmits the trajectories to the followers according to the proposed network structure (Figure 1a). Then, each follower tracks its reference trajectory using the MPC.

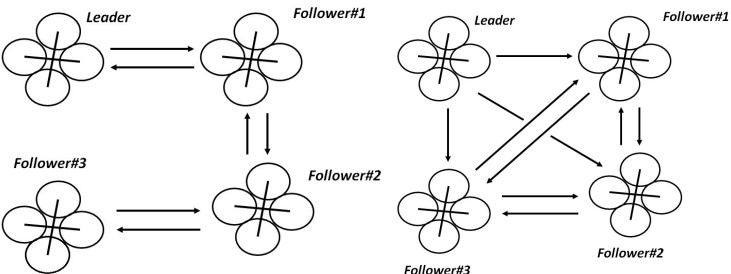

**Figure 1.** Network structure comparison: (**a**) network structure adopted in Reference [13]; (**b**) network structure adopted in this study.

In this study, the consensus-based algorithm is moved into the follower's formation strategy in order to minimize the data that are transmitted around the network. The distributed network structure in Figure 1b is considered. In the horizontal motion, the leader tracks the given waypoints by using the MPC, and generates the desired formation trajectory of each follower based on its flight information, predicted trajectory, and the given formation pattern. The leader transmits the desired formation trajectories to the followers but does not receive any data from the followers. Moreover, the desired formation trajectory of each follower contains only the information of where the follower should be, that is, the desired position over the predicted time horizon. In the vertical motion, the leader tracks

the desired altitude and climb rate by using the MPC, and generates the desired formation trajectory for each follower. Likewise, the desired formation trajectory for each follower in the vertical motion only contains the information of the desired altitude.

The follower's formation control in the horizontal motion is carried out based on the distributed consensus control and the MPC. Consensus control requires the information of the desired velocity, attitude, and angular velocity. However, the desired formation trajectory of each follower generated by the leader does not contain this information. Therefore, an algorithm using a finite difference method is added to the formation strategy in the horizontal motion to generate the desired information from the desired formation trajectory required by the consensus control. Once the complete desired formation trajectory is generated, the formation control engages. The consensus control generates the flight paths of the follower under consideration and the other followers that are connected to it in the proposed network structure. However, the flight paths of other followers are not used by the proposed formation strategy of the follower under consideration. Despite that these paths might be valuable in cases such as interruption of communication or a collision-avoidance situation, they are beyond the scope of this study. The flight path of the follower under consideration is passed to the MPC. The MPC tracks the flight path and computes control inputs based on the unconstrained QP problem derived from the LQT. In the vertical motion, the LQT requires the information of the desired vertical velocity that is not included in the desired formation trajectory generated by the leader. Likewise, an algorithm using a finite-difference method is added to the formation strategy in the vertical motion to generate the desired information from the desired formation trajectory required by the LQT. Then, the LQT can compute the formation control based on the complete desired formation trajectory.

## 3. Dynamic Model of a Quadrotor

### 3.1. EOM of the Quadrotor

A quadrotor's dynamics considering a flat Earth with an atmosphere at rest can be described by Newton–Euler formalism. The reference frames of the quadrotor are defined before EOMs are derived. The configuration, forces, and reference frames of the quadrotor that is considered in this study are illustrated in Figure 2.

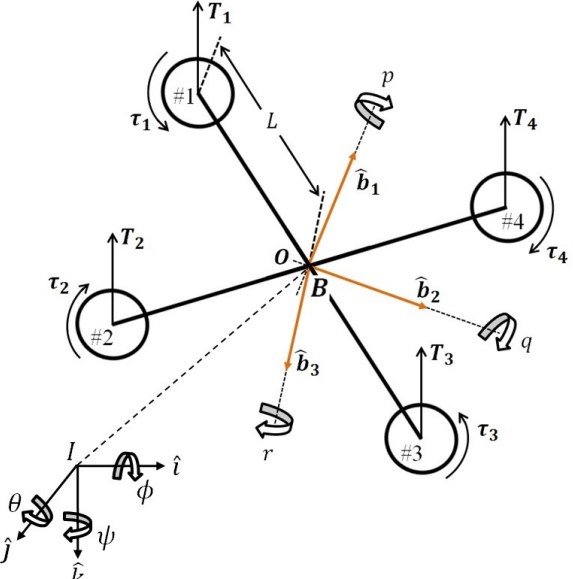

**Figure 2.** Thrusts, torques, quadrotor's parameter, and reference frames.

In Figure 2, $I = \left(\hat{i}, \hat{j}, \hat{k}\right)$ denotes the inertial frame and $B = \left(\hat{b}_1, \hat{b}_2, \hat{b}_3\right)$ denotes the body-fixed frame of the quadrotor. The axes $\left(\hat{i}, \hat{j}, \hat{k}\right)$ of the frame $I$ point to North, East, and downward under the consideration of flat Earth, respectively. The origin of the body-fixed frame $B$ coincides with the center of gravity (CG) of the quadrotor at $O$. In addition, $L$ represents the arm length of the rotor measured from $O$ to the rotation axis. The weight of the quadrotor is not shown in Figure 2, but it should be considered while deriving the EOMs. $T_i$ and $\boldsymbol{\tau}_i$ are the thrust and torque of i-th rotor ($i = 1, 2, 3, 4$) with respect to the frame $B$, respectively. $T_i$ and $\boldsymbol{\tau}_i$ can be written as follows:

$$T_i = -T_i \hat{b}_3, \tag{1}$$

$$\boldsymbol{\tau}_i = (-1)^i \tau_i \hat{b}_3. \tag{2}$$

The Newton–Euler formalism with respect to frame $B$ is shown as follows:

$$\begin{bmatrix} m I_3 & \mathbf{0}_{3\times3} \\ \mathbf{0}_{3\times3} & I_J \end{bmatrix} \begin{bmatrix} \dot{V}_B \\ \dot{\omega}_B \end{bmatrix} + \begin{bmatrix} \omega_B \times (m V_B) \\ \omega_B \times (I_J \omega_B) \end{bmatrix} = \begin{bmatrix} F_B \\ \tau_B \end{bmatrix}, \tag{3}$$

$$I_J = \begin{bmatrix} I_{xx} & I_{xy} & I_{xz} \\ I_{yx} & I_{yy} & I_{yz} \\ I_{zx} & I_{zy} & I_{zz} \end{bmatrix}. \tag{4}$$

In Equation (3), $m$ is the mass of the quadrotor and $I_3, \mathbf{0}_{3\times3} \in \mathbb{R}^{3\times3}$ denote the identity matrix and square zero matrix, respectively. $I_J$ is the inertial tensor of the quadrotor. The quadrotor is symmetric to the xz and yz planes. Therefore, $I_{xy} = I_{yx} = I_{yz} = I_{zy} = 0$. Furthermore, $I_{zx}$ and $I_{xz}$ are relatively small compared to $I_{xx}$, $I_{yy}$, and $I_{zz}$. Hence, the inertial tensor of the quadrotor becomes $I_J = diag(I_{xx}, I_{yy}, I_{zz})$. $V_B = (u, v, w)$ and $\omega_B = (p, q, r)$ are the velocity and angular velocity of the quadrotor with respect to frame $B$. $F_B$ are the forces acting on the quadrotor with respect to frame $B$, including the thrust of each rotor and the weight of the quadrotor. $\tau_B$ includes the torque induced by each rotor and the moment induced by the thrust of each rotor.

To obtain the overall EOMs, the transformation between the velocity in frame $I$ and frame $B$, and the transformation between the angular velocity and the rate change of the Euler angles must be included. The above transformations are commonly seen in the derivation of the EOMs of unmanned aerial vehicles (UAVs) and drones. More details can be found in [14,15]. Here, rotation matrix $R_{I2B}$, which transforms the coordinate of a vector from frame $I$ to frame $B$, is introduced. Rotation matrix $R_{I2B}$ is defined as follows:

$$R_{I2B} = \begin{bmatrix} C_\theta C_\psi & C_\theta S_\psi & -S_\theta \\ S_\phi S_\theta C_\psi - C_\phi S_\psi & S_\phi S_\theta S_\psi + C_\phi C_\psi & S_\phi C_\theta \\ C_\phi S_\theta C_\psi + S_\phi S_\psi & C_\phi S_\theta S_\psi - S_\phi C_\psi & C_\phi C_\theta \end{bmatrix}, \tag{5}$$

where $S_{(\cdot)} = \sin(\cdot)$ and $C_{(\cdot)} = \cos(\cdot)$. In addition, $R_{I2B}$ is an orthogonal matrix. Therefore, the following property holds:

$$R_{I2B}^{-1} = R_{I2B}^{T} = R_{B2I}. \tag{6}$$

The transformation between the velocity with respect to frame $I$, $V_I = (\dot{x}, \dot{y}, \dot{z})$, and the velocity with respect to frame $B$, $V_B = (u, v, w)$, can be written as follows:

$$V_I = R_{B2I} V_B, \tag{7}$$

$$\Rightarrow \begin{bmatrix} \dot{x} \\ \dot{y} \\ \dot{z} \end{bmatrix} = \begin{bmatrix} C_\theta C_\psi & S_\phi S_\theta C_\psi - C_\phi S_\psi & C_\phi S_\theta C_\psi + S_\phi S_\psi \\ C_\theta S_\psi & S_\phi S_\theta S_\psi + C_\phi C_\psi & C_\phi S_\theta S_\psi - S_\phi C_\psi \\ -S_\theta & S_\phi C_\theta & C_\phi C_\theta \end{bmatrix} \begin{bmatrix} u \\ v \\ w \end{bmatrix}, \tag{8}$$

where $(x, y, z)$ are the position of the quadrotor with respect to frame $\mathbf{I}$. Additionally, the transformation between the angular velocity and the rate change of the Euler angles is shown as follows:

$$\begin{bmatrix} \dot{\phi} \\ \dot{\theta} \\ \dot{\psi} \end{bmatrix} = \begin{bmatrix} 1 & S_\phi \tan\theta & C_\phi \tan\theta \\ 0 & C_\phi & -S_\phi \\ 0 & S_\phi \sec\theta & C_\phi \sec\theta \end{bmatrix} \begin{bmatrix} p \\ q \\ r \end{bmatrix}. \tag{9}$$

Finally, the EOMs of the quadrotor can be obtained with the above definitions and transformations. In this study, the formation control is decoupled into horizontal and vertical motions. Therefore, the EOMs of the quadrotor are arranged in a similar form.

In the horizontal motion, states $(x, y, u, v, \theta, \phi, q, p)$ are considered, and the EOMs are arranged as follows:

$$Pitch\ Axis: \begin{bmatrix} \dot{x} \\ \dot{u} \\ \dot{\theta} \\ \dot{q} \end{bmatrix} = \begin{bmatrix} (C_\theta C_\psi)u + (S_\phi S_\theta C_\psi - C_\phi S_\psi)v + (C_\phi S_\theta C_\psi + S_\phi S_\psi)w \\ rv - qw - g\sin\theta \\ qC_\phi - rS_\phi \\ \frac{rp(I_{zz} - I_{xx})}{I_{yy}} + \frac{1}{I_{yy}}M_\theta \end{bmatrix},$$

$$\tag{10}$$

$$Roll\ Axis: \begin{bmatrix} \dot{y} \\ \dot{v} \\ \dot{\phi} \\ \dot{p} \end{bmatrix} = \begin{bmatrix} (C_\theta S_\psi)u + (S_\phi S_\theta S_\psi + C_\phi C_\psi)v + (C_\phi S_\theta S_\psi - S_\phi C_\psi)w \\ pw - ru + g\sin\phi\cos\theta \\ p + qS_\phi \tan\theta + rC_\phi \tan\theta \\ \frac{qr(I_{yy} - I_{zz})}{I_{xx}} + \frac{1}{I_{xx}}M_\phi \end{bmatrix}.$$

On the other hand, the EOMs considering the states $(z, w, \psi, r)$ in the vertical motion are arranged as follows:

$$\begin{bmatrix} \dot{z} \\ \dot{w} \\ \dot{\psi} \\ \dot{r} \end{bmatrix} = \begin{bmatrix} (-S_\theta)u + (S_\phi C_\theta)v + (C_\phi C_\theta)w \\ qu - pv + g\cos\phi\cos\theta - \frac{1}{m}T_{total} \\ qS_\phi \sec\theta + rC_\phi \sec\theta \\ \frac{pq(I_{xx} - I_{yy})}{I_{zz}} + \frac{1}{I_{zz}}M_\psi \end{bmatrix}. \tag{11}$$

In Equations (10) and (11), $M_\theta$, $M_\phi$, $M_\psi$, and $T_{total}$ are the controls and defined as follows:

$$\begin{aligned} M_\phi &= \tfrac{\sqrt{2}}{2}L(T_1 + T_2 - T_3 - T_4), \\ M_\theta &= \tfrac{\sqrt{2}}{2}L(T_1 - T_2 - T_3 + T_4), \\ M_\psi &= -\tau_1 + \tau_2 - \tau_3 + \tau_4, \\ T_{total} &= T_1 + T_2 + T_3 + T_4. \end{aligned} \tag{12}$$

### 3.2. Linearization of the EOMs

The quadrotor model that is used in the consensus control and MPC is obtained by linearizing Equations (10) and (11). The small disturbance method is considered to linearize the EOMs, and the hovering condition of the quadrotor is chosen as the equilibrium reference. The equilibrium point of each state in horizontal and vertical motions are listed in Tables 1 and 2, respectively.

**Table 1.** Equilibrium pt. and small disturbance expression of each state and controls in horizontal motion.

| State | Equilibrium pt. | Small Disturbance Expression | Unit |
|---|---|---|---|
| $x$ | $x_0 = x_{Init}$ | $\Delta x$ | m |
| $y$ | $y_0 = y_{Init}$ | $\Delta y$ | m |
| $u$ | $u_0 = 0$ | $\Delta u$ | m/s |
| $v$ | $v_0 = 0$ | $\Delta v$ | m/s |
| $\theta$ | $\theta_0 = 0$ | $\Delta \theta$ | rad |
| $\phi$ | $\phi_0 = 0$ | $\Delta \phi$ | rad |
| $q$ | $q_0 = 0$ | $\Delta q$ | rad/s |
| $p$ | $p_0 = 0$ | $\Delta p$ | rad/s |
| $M_\theta$ | $M_{\theta,0} = 0$ | $\Delta M_\theta$ | m · N |
| $M_\phi$ | $M_{\phi,0} = 0$ | $\Delta M_\phi$ | m · N |

**Table 2.** Equilibrium pt. and small disturbance expression of each state and controls in vertical motion.

| State | Equilibrium pt. | Small Disturbance Expression | Unit |
|---|---|---|---|
| $z$ | $z_0 = z_{Init}$ | $\Delta z$ | m |
| $w$ | $w_0 = 0$ | $\Delta w$ | m/s |
| $\psi$ | $\psi_0 = 0$ | $\Delta \psi$ | rad |
| $r$ | $r_0 = 0$ | $\Delta r$ | rad/s |
| $M_\psi$ | $M_{\psi,0} = 0$ | $\Delta M_\psi$ | m · N |
| $T_{total}$ | $T_{total,0} = mg$ | $\Delta T_{total}$ | N |

where $x_{Init}$, $y_{Init}$, and $z_{Init}$ are the initial position of the quadrotor. Subscript 0 denotes the equilibrium point of the state, and $\Delta(\cdot)$ denotes the small disturbance of the state.

Replacing the states in Equations (10) and (11) by the sum of the equilibrium point and the small disturbance of the states, and ignoring the cross-product terms of the small disturbance of the states, the linearized EOMs of the quadrotor in the horizontal and vertical motions can be obtained as follows:

$$
\begin{aligned}
Pitch\ Axis: \begin{bmatrix} \Delta \dot{x} \\ \Delta \dot{u} \\ \Delta \dot{\theta} \\ \Delta \dot{q} \end{bmatrix} &= \begin{bmatrix} 0 & 1 & 0 & 0 \\ 0 & 0 & -g & 0 \\ 0 & 0 & 0 & 1 \\ 0 & 0 & 0 & 0 \end{bmatrix} \begin{bmatrix} \Delta x \\ \Delta u \\ \Delta \theta \\ \Delta q \end{bmatrix} + \begin{bmatrix} 0 \\ 0 \\ 0 \\ \frac{1}{I_{yy}} \end{bmatrix} \Delta M_\theta, \\[2em]
Roll\ Axis: \begin{bmatrix} \Delta \dot{y} \\ \Delta \dot{v} \\ \Delta \dot{\phi} \\ \Delta \dot{p} \end{bmatrix} &= \begin{bmatrix} 0 & 1 & 0 & 0 \\ 0 & 0 & g & 0 \\ 0 & 0 & 0 & 1 \\ 0 & 0 & 0 & 0 \end{bmatrix} \begin{bmatrix} \Delta y \\ \Delta v \\ \Delta \phi \\ \Delta p \end{bmatrix} + \begin{bmatrix} 0 \\ 0 \\ 0 \\ \frac{1}{I_{xx}} \end{bmatrix} \Delta M_\phi,
\end{aligned}
\tag{13}
$$

$$
\begin{bmatrix} \Delta \dot{z} \\ \Delta \dot{w} \\ \Delta \dot{\psi} \\ \Delta \dot{r} \end{bmatrix} = \begin{bmatrix} 0 & 1 & 0 & 0 \\ 0 & 0 & 0 & 0 \\ 0 & 0 & 0 & 1 \\ 0 & 0 & 0 & 0 \end{bmatrix} \begin{bmatrix} \Delta z \\ \Delta w \\ \Delta \psi \\ \Delta r \end{bmatrix} + \begin{bmatrix} 0 & 0 \\ \frac{-1}{m} & 0 \\ 0 & 0 \\ 0 & \frac{1}{I_{zz}} \end{bmatrix} \begin{bmatrix} \Delta T_{total} \\ \Delta M_\psi \end{bmatrix}.
\tag{14}
$$

## 4. Control Law Preliminaries

### 4.1. Consensus Control

The main objective of the consensus control in this study is to generate the flight path for the MPC in the follower's horizontal formation strategy. The consensus control of an agent is generated by taking information such as the flight states of other connected agents in the network structure into consideration, and eventually all the agents reach a consensus state through the consensus control.

Consider the leader–follower architecture with one leader and $N_F$ followers; consensus control input $u_i(k)$ of the i-th follower at time step $k$ is given as follows:

$$\boldsymbol{u}_i(k) \quad = \quad -\sum_{j=1}^{N_F+1} a_{ij} \left[ \sum_{ks=1}^{Ns} \beta_{ks} \left( \hat{\boldsymbol{s}}_i^{(ks)}(k) - \hat{\boldsymbol{s}}_j^{(ks)}(k) \right) \right], \tag{15}$$

$$\hat{\boldsymbol{s}}_i^{(ks)}(k) \quad = \quad \boldsymbol{s}_i^{(ks)}(k) - \boldsymbol{d}_i^{(ks)}(k),$$

$$\hat{\boldsymbol{s}}_j^{(ks)}(k) \quad = \quad \boldsymbol{s}_j^{(ks)}(k) - \boldsymbol{d}_j^{(ks)}(k),$$

where subscript *j* denotes the j-th follower in the network structure other than i-th follower. As in the case that subscript $j = N_F + 1$, it refers to the leader. $a_{ij}$ represents the communication status between the i-th and j-th followers. If the i-th follower is receiving data from the j-th follower or leader $(j = N_F + 1)$, then $a_{ij} = 1$. Otherwise, $a_{ij} = 0$. $\beta_{ks}$ $(ks = 1, 2, ..., Ns)$ are the control gains, where $Ns$ is the total number of the states. $\hat{\boldsymbol{s}}_{(\cdot)}^{(ks)}(k)$ denotes the state error between the state $\boldsymbol{s}_{(\cdot)}^{(ks)}(k)$ and reference state $\boldsymbol{d}_{(\cdot)}^{(ks)}(k)$ at time step *k*.

### 4.2. Model Predictive Control

Model predictive control is a control algorithm based on the optimal theory and the plant model. In MPC, it first obtains the optimal trajectory and control sequence by solving the optimal problem according to the given cost function. The optimal trajectory can be seen as the prediction of the plant behavior over the predicted time horizon with the optimal control sequence. However, the plant might not behave identically to the prediction. Thus, MPC applies only to the first control in the optimal control sequence to the plant. When the next measurements of the states are acquired, MPC repeats the above sequence. By doing so, the MPC algorithm works like an optimal feedback control.

The following sections show the problem statement, the solution to the LQT problem, and how to rewrite an LQT problem into a QP problem.

#### 4.2.1. Linear Quadratic Tracker

This section shows the problem statement and the solution to the LQT. The derivation of the solution to the LQT is not covered in this section. If the readers are interested in the derivation, it can be found in Reference [16].

Consider the following discrete state-space model with *n* states, *m* controls, and *q* outputs:

$$\begin{aligned} \boldsymbol{x}_{k+1} \quad &= \quad A\boldsymbol{x}_k + B\boldsymbol{u}_k, \\ \boldsymbol{y}_k \quad &= \quad C\boldsymbol{x}_k, \end{aligned} \tag{16}$$

where the system matrix $A \in \mathbb{R}^{n \times n}$, the control matrix $B \in \mathbb{R}^{n \times m}$, and the output matrix $C \in \mathbb{R}^{q \times n}$. $\boldsymbol{x}_k \in \mathbb{R}^n$ is a column vector, which denotes the state at time step *k* in the considered time horizon. $\boldsymbol{u}_k \in \mathbb{R}^m$ is a column vector, which denotes the control at time step *k* in the considered time horizon. $\boldsymbol{y}_k \in \mathbb{R}^q$ is a column vector, which denotes the output state at time step *k* in the considered time horizon. Next, define the linear quadratic cost function based on the control and output state in the discrete state-space model and the reference trajectory for the output state as follows:

$$J = \frac{1}{2}(\boldsymbol{y}_{N_p} - \boldsymbol{r}_{N_p})^T P(\boldsymbol{y}_{N_p} - \boldsymbol{r}_{N_p}) + \frac{1}{2} \sum_{k=0}^{N_p-1} \left[ (\boldsymbol{y}_k - \boldsymbol{r}_k)^T Q(\boldsymbol{y}_k - \boldsymbol{r}_k) + \boldsymbol{u}_K^T R \boldsymbol{u}_k \right], \tag{17}$$

where $k$ $(k = 0, 1, ..., N_p)$ denotes the time step over the time horizon. $\boldsymbol{r}_k$ $(k = 0, 1, ..., N_p)$ is a column vector, which denotes the reference trajectory for the output state at time step *k*. $P, Q, R$ are the weighting matrices. All the weighting matrices are symmetric. $P, Q \in \mathbb{R}^{q \times q}$ are positive semidefinite. $R \in \mathbb{R}^{m \times m}$ is positive definite:

$$P \geq 0, \quad Q \geq 0, \quad R > 0. \tag{18}$$

The optimized control sequence $u_k$ ($k = 0, 1, ..., N_p - 1$) that minimizes the cost function $J$ in Equation (17) can be obtained by solving the following equations:

$$
\begin{aligned}
K_k &= (B^T S_{k+1} B + R)^{-1} B^T S_{k+1} A, \quad S_{N_p} = C^T P C, \\
S_k &= A^T S_{k+1} (A - BK_k) + C^T Q C, \\
v_k &= (A - BK_k)^T v_{k+1} + C^T Q r_k, \quad v_{N_p} = C^T P r_{N_p}, \\
K_k^v &= (B^T S_{k+1} B + R)^{-1} B^T, \\
u_k &= -K_k x_k + K_k^v v_{k+1}.
\end{aligned}
\tag{19}
$$

In Equation (19), $K_k$, $S_k$, and $K_k^v$ can be solved offline since they only consider the matrices in the discrete state-space model (16) and the weighting matrices $P$, $Q$, and $R$ given by the users. Furthermore, by solving $v_k$ backward in time, the optimal tracking control sequence can be obtained. Substituting the optimal control sequence into the discrete state-space model (16), the optimal tracking trajectory over the time horizon can be obtained.

### 4.2.2. Unconstrained Quadratic Programming Problem

The quadratic programming problem deals with the quadratic cost function that is subject to linear equality and/or inequality constraints on the variables in the cost function. A quadratic programming problem can be stated as follows:

$$
\begin{aligned}
minimize \quad J_{QP} &= \frac{1}{2} \xi^T Q_c \xi + c^T \xi, \\
subject\ to \quad A_c \xi &\leq b, \\
E\xi &= d,
\end{aligned}
\tag{20}
$$

where $J_{QP}$ denotes the cost function of the quadratic programming problem. $\xi \in \mathbb{R}^n$ is a column vector that contains $n$ variables to be optimized. $Q_c \in \mathbb{R}^{n \times n}$ is a real symmetric matrix. $c \in \mathbb{R}^n$ is a real-valued column vector. $A_c \in \mathbb{R}^{m \times n}$ and $b \in \mathbb{R}^m$ describe the inequality constraints of the variables. Similarly, $E \in \mathbb{R}^{m \times n}$ and $d \in \mathbb{R}^m$ describe the equality constraints of the variables.

To find an analytical solution to a constrained quadratic programming problem is difficult, if not impossible. However, it can still be solved by methods such as interior-point, active-set, or gradient-projection. On the contrary, the optimized solution to the unconstrained quadratic programming problem can be found simply by solving the linear equations of the partial derivative of the modified cost function with respect to each variable and set to zero.

In formation control, our objective is to find the control input that tracks the reference trajectory. By rewriting the LQT into a quadratic programming problem, the variables to be optimized become the control input sequence. That is, we can obtain the optimal control sequence by solving linear equations directly.

To rewrite the cost function, we start from rearranging the discrete state-space model (16). Expanding the discrete state-space model with respect to time step $k$ ($k = 0, 1, ..., N_p - 1$):

$$
\begin{aligned}
x_1 &= Ax_0 + Bu_0, \\
x_2 &= Ax_1 + Bu_1 = A(Ax_0 + B_0) + Bu_1 = A^2 x_0 + ABu_0 + Bu_1, \\
x_3 &= Ax_2 + Bu_2 = A\left(A^2 x_0 + ABu_0 + Bu_1\right) + Bu_2 = A^3 x_0 + A^2 Bu_0 + ABu_1 + Bu_2, \\
&\vdots \\
x_{N_p} &= Ax_{N_p-1} + Bu_{N_p-1} = A^{N_p} x_0 + A^{N_p-1} Bu_0 + \cdots + Bu_{N_p-1},
\end{aligned}
\tag{21}
$$

and defining the following vectors:

$$
\begin{aligned}
\widetilde{x} &= \left[\begin{array}{cccc} x_1^T & x_2^T & \cdots & x_{N_P}^T \end{array}\right]^T, \\
\widetilde{u} &= \left[\begin{array}{cccc} u_0^T & u_1^T & \cdots & u_{N_p-1}^T \end{array}\right]^T,
\end{aligned}
\tag{22}
$$

where $\widetilde{x} \in \mathbb{R}^{nN_p \times 1}$ and $\widetilde{u} \in \mathbb{R}^{mN_p \times 1}$. Equation (21) can be rewritten into the matrix form expressed by the initial condition and the vectors in Equation (22) as follows:

$$
\begin{aligned}
\widetilde{x} &= \begin{bmatrix} A \\ A^2 \\ A^3 \\ \vdots \\ A^{N_p} \end{bmatrix} x_0 + \begin{bmatrix} B & 0_{n\times m} & \cdots & 0_{n\times m} \\ AB & B & \cdots & \vdots \\ \vdots & \vdots & \ddots & 0_{n\times m} \\ A^{N_p-1}B & A^{N_p-2}B & \cdots & B \end{bmatrix} \widetilde{u} \\
&= \widetilde{A}x_0 + \widetilde{B}\widetilde{u},
\end{aligned}
\tag{23}
$$

where $\widetilde{A} \in \mathbb{R}^{nN_p \times n}$ and $\widetilde{B} \in \mathbb{R}^{nN_p \times mN_p}$. Before substituting Equation (23) into the cost function (17), substitute the output function in Equation (16) into cost function (17) and expand the cost function:

$$
\begin{aligned}
J &= \frac{1}{2}\left(y_{N_p} - r_{N_p}\right)^T P\left(y_{N_p} - r_{N_p}\right) + \frac{1}{2}\sum_{k=0}^{N_p-1}\left[(y_k - r_k)^T Q\,(y_k - r_k) + u_K^T R u_k\right] \\
&= \frac{1}{2}\left(Cx_{N_p} - r_{N_p}\right)^T P\left(Cx_{N_p} - r_{N_p}\right) + \frac{1}{2}\sum_{k=0}^{N_p-1}\left[(Cx_k - r_k)^T Q\,(Cx_k - r_k) + u_K^T R u_k\right] \\
&= \frac{1}{2}\left(x_{N_p}^T C^T P C x_{N_p} - 2x_{N_p}^T C^T P r_{N_p} + r_{N_p}^T P r_{N_p}\right) + \cdots \\
&\qquad\qquad \frac{1}{2}\sum_{k=0}^{N_p-1}\left[x_k^T C^T Q C x_k - 2x_k^T C^T Q r_k + r_k^T Q r_k + u_k^T R u_k\right] \\
&= \frac{1}{2}\widetilde{x}^T \begin{bmatrix} C^T Q C & 0_{n\times n} & \cdots & 0_{n\times n} \\ 0_{n\times n} & \ddots & & \vdots \\ \vdots & & C^T Q C & 0_{n\times n} \\ 0_{n\times n} & \cdots & 0_{n\times n} & C^T P C \end{bmatrix}\widetilde{x} - \widetilde{x}^T \begin{bmatrix} C^T Q & 0_{n\times q} & \cdots & 0_{n\times q} \\ 0_{n\times q} & \ddots & & \vdots \\ \vdots & & C^T Q & 0_{n\times q} \\ 0_{n\times q} & \cdots & 0_{n\times q} & C^T P \end{bmatrix}\widetilde{r} + \cdots \\
&\qquad \widetilde{u}^T \begin{bmatrix} R & 0_{m\times m} & \cdots & 0_{m\times m} \\ 0_{m\times m} & \ddots & & \vdots \\ \vdots & & \ddots & 0_{m\times m} \\ 0_{m\times m} & \cdots & 0_{m\times m} & R \end{bmatrix}\widetilde{u} + \widetilde{r}^T \begin{bmatrix} Q & 0_{q\times q} & \cdots & 0_{q\times q} \\ 0_{q\times q} & \ddots & & \vdots \\ \vdots & & Q & 0_{q\times q} \\ 0_{q\times q} & \cdots & 0_{q\times q} & P \end{bmatrix}\widetilde{r} + \cdots \\
&\qquad x_0 C^T Q C x_0 - 2x_0 C^T Q r_0 + r_0^T Q r_0 \\
&= \frac{1}{2}\widetilde{x}^T \widetilde{Q}_1 \widetilde{x} - \widetilde{x}^T \widetilde{Q}_2 \widetilde{r} + \frac{1}{2}\widetilde{u}^T \widetilde{R}\widetilde{u} + \underbrace{\frac{1}{2}\widetilde{r}^T \widetilde{Q}_3 \widetilde{r} + x_0 C^T Q C x_0 - 2x_0 C^T Q r_0 + r_0^T Q r_0}_{\textit{Constant contribution to cost function}},
\end{aligned}
\tag{24}
$$

where $\widetilde{Q}_1 \in \mathbb{R}^{nN_p \times nN_p}$, $\widetilde{Q}_2 \in \mathbb{R}^{nN_p \times qN_p}$, $\widetilde{Q}_3 \in \mathbb{R}^{qN_p \times qN_p}$, $\widetilde{r} \in \mathbb{R}^{qN_p \times 1}$ and $\widetilde{R} \in \mathbb{R}^{mN_p \times mN_p}$. In Equation (24), $x_0$ is the given initial condition and $r_0$ is the reference of the output state at $k = 0$. $\widetilde{r}$ is a column vector consists of the given reference trajectory of the output state at each time step. Thus, the terms depend only on $x_0$, $r_0$, or $\widetilde{r}$ are constant contribution to cost function $J$ and can be ignored in the subsequent derivation. The modified cost function now becomes:

$$
\widetilde{J} = \frac{1}{2}\widetilde{x}^T \widetilde{Q}_1 \widetilde{x} - \widetilde{x}^T \widetilde{Q}_2 \widetilde{r} + \frac{1}{2}\widetilde{u}^T \widetilde{R}\widetilde{u}.
\tag{25}
$$

Replacing $\widetilde{x}$ in Equation (25) by Equation (23)

$$
\begin{aligned}
\widetilde{J} &= \frac{1}{2}\left(\widetilde{A}x_0 + \widetilde{B}\widetilde{u}\right)^T \widetilde{Q}_1 \left(\widetilde{A}x_0 + \widetilde{B}\widetilde{u}\right) - \left(\widetilde{A}x_0 + \widetilde{B}\widetilde{u}\right)^T \widetilde{Q}_2 \widetilde{r} + \frac{1}{2}\widetilde{u}^T \widetilde{R}\widetilde{u} \\
&= \frac{1}{2}\widetilde{u}^T \left(\widetilde{B}^T \widetilde{Q}_1 \widetilde{B} + \widetilde{R}\right)\widetilde{u} + \begin{bmatrix} x_0^T & \widetilde{r}^T \end{bmatrix} \begin{bmatrix} \widetilde{A}^T \widetilde{Q}_1 \widetilde{B} \\ -\widetilde{Q}_2^T \widetilde{B} \end{bmatrix}\widetilde{u} + \underbrace{\frac{1}{2}x_0^T \widetilde{A}^T \widetilde{Q}_1 \widetilde{A}x_0 - x_0^T \widetilde{A}^T \widetilde{Q}_2 \widetilde{r}}_{Constant\ contribution\ to\ cost\ function}
\end{aligned}
\tag{26}
$$

and ignoring the constant contribution part in cost function $\widetilde{J}$, the cost function becomes

$$
\begin{aligned}
\overline{J} &= \frac{1}{2}\widetilde{u}^T \left(\widetilde{B}^T \widetilde{Q}_1 \widetilde{B} + \widetilde{R}\right)\widetilde{u} + \begin{bmatrix} x_0^T & \widetilde{r}^T \end{bmatrix}\begin{bmatrix} \widetilde{A}^T \widetilde{Q}_1 \widetilde{B} \\ -\widetilde{Q}_2^T \widetilde{B} \end{bmatrix}\widetilde{u} \\
&= \frac{1}{2}\widetilde{u}^T H\widetilde{u} + \overline{x}F\widetilde{u},
\end{aligned}
\tag{27}
$$

where $H \in \mathbb{R}^{mN_p \times mN_p}$, $\overline{x} \in \mathbb{R}^{(n+qN_p)}$ is a row vector consists of initial condition $x_0$ and the reference trajectory at each time step and $F \in \mathbb{R}^{(n+qN_p) \times mN_p}$. The optimal control sequence can be obtained by taking the partial derivative of $\overline{J}$ with respect to $\widetilde{u}$ and setting it to zero:

$$
\begin{aligned}
\frac{\partial \overline{J}}{\partial \widetilde{u}} &= \frac{1}{2}\left(H\widetilde{u} + H^T \widetilde{u}\right) + \left(\overline{x}F\right)^T = 0 \\
&= H\widetilde{u} + \left(\overline{x}F\right)^T = 0.
\end{aligned}
\tag{28}
$$

The optimal control sequence is then given by

$$
\widetilde{u} = -H^{-1}F^T \overline{x}^T.
\tag{29}
$$

Equation (29) will be used in the MPC in leader's and follower's horizontal formation strategies to compute the optimal tracking control sequence.

## 5. Formation Strategies

The formation control strategies of the leader and the followers are discussed in detail in the following sections. How the leader generates followers' desired formation trajectories in both horizontal and vertical motions will be carried out in Section 5.1. Section 5.2 shows how a follower computes the formation control input using consensus control and MPC after receiving the desired formation trajectory information.

### 5.1. Formation Strategy: Leader

Figure 3 shows the block diagram of a leader's formation strategy. First, the leader tracks the predefined waypoints and desired altitude/climb rate using the MPC. The MPC is formulated as a QP problem for the horizontal motion and LQT for the vertical motion. After the predicted trajectory is generated along with the tracking control input, a built-in algorithm generates followers' desired formation trajectories and the trajectories are transmitted to each follower.

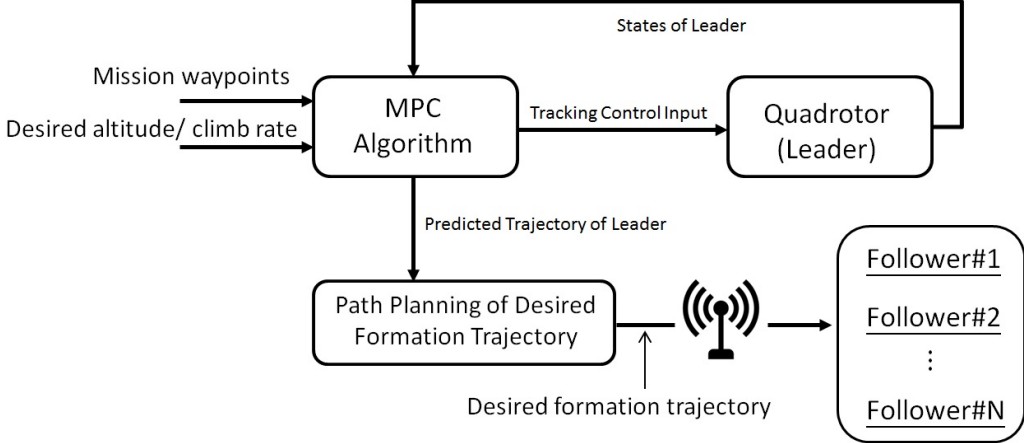

**Figure 3.** Formation control strategy of the leader.

### 5.1.1. Generation of the Desired Formation Trajectory in the Horizontal Motion

In the horizontal motion, the leader generates the followers' desired formation trajectories based on its position, direction of the horizontal velocity in frame *I*, predicted trajectory, and the given formation pattern. The transformation from the leader's predicted trajectory to the desired formation trajectories of the followers can be derived according to Figure 4.

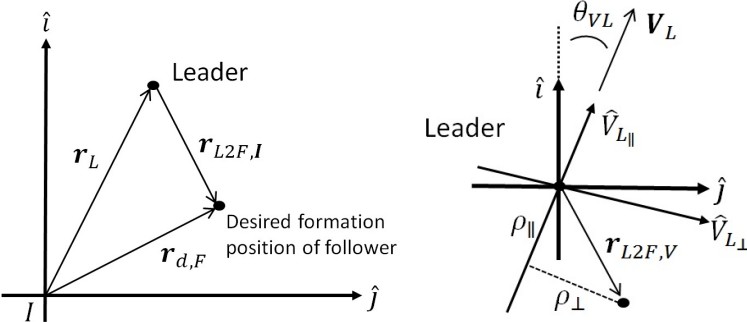

**Figure 4.** Relative position of the leader and the follower (left); tangential coordinate and leader's velocity $V_L$ in a horizontal plane (right).

First of all, from Figure 4a, the follower's desired formation position in horizontal plane in frame *I* can be described as follows:

$$\boldsymbol{r}_{d,F} = \boldsymbol{r}_L + \boldsymbol{r}_{L2F,I}, \tag{30}$$

where $\boldsymbol{r}_{d,F}$ denotes the position vector of the follower's desired formation position and $\boldsymbol{r}_L$ denotes the leader's position vector. $\boldsymbol{r}_{L2F,I}$ is a vector pointing from leader's position to follower's desired formation position with respect to frame *I*.

Considering Figure 4b, the desired formation position with respect to the leader's velocity tangential coordinate can be written as follows:

$$\boldsymbol{r}_{L2F,V} = \begin{bmatrix} \rho_\parallel & \rho_\perp \end{bmatrix} \begin{bmatrix} \hat{V}_{L_\parallel} \\ \hat{V}_{L_\perp} \end{bmatrix}, \tag{31}$$

where $\hat{V}_{L_\parallel}$ and $\hat{V}_{L_\perp}$ are the unit vectors of the leader's velocity tangential coordinate in which $\hat{V}_{L_\parallel}$ is parallel to $V_L$ and $\hat{V}_{L_\perp}$ is perpendicular to $V_L$. $\left( \rho_\parallel, \rho_\perp \right)$ is the coordinate of the desired formation position with respect to the leader's tangential coordinate.

The rotation matrix, denoted as $R_{V2I}$, that transforms the position vector from $r_{L2F,V}$ to $r_{L2F,I}$ is defined as follows:

$$R_{V2I} = \begin{bmatrix} \cos\theta_{VL} & \sin\theta_{VL} \\ -\sin\theta_{VL} & \cos\theta_{VL} \end{bmatrix}, \tag{32}$$

where $\theta_{VL}$ is the angle between $\boldsymbol{V}_L$ and $\hat{\boldsymbol{i}}$, which can be obtained by

$$\theta_{VL} = \cos^{-1}\left(\frac{\boldsymbol{V}_L \cdot \hat{\boldsymbol{i}}}{\|\boldsymbol{V}_L\|}\right). \tag{33}$$

$\boldsymbol{V}_L$ and its magnitude can be computed as follows:

$$\begin{aligned} \boldsymbol{V}_L &= \dot{x}\,\hat{\boldsymbol{i}} + \dot{y}\,\hat{\boldsymbol{j}}, \\ \|\boldsymbol{V}_L\| &= \sqrt{\dot{x}^2 + \dot{y}^2}, \end{aligned} \tag{34}$$

where $\dot{x}$ and $\dot{y}$ can be obtained from Equation (8). Thus, position vector $r_{L2F,I}$ can be obtained as follows:

$$r_{L2F,I} = \begin{bmatrix} \rho_\parallel & \rho_\perp \end{bmatrix} \begin{bmatrix} \cos\theta_{VL} & \sin\theta_{VL} \\ -\sin\theta_{VL} & \cos\theta_{VL} \end{bmatrix} \begin{bmatrix} \hat{\boldsymbol{i}} \\ \hat{\boldsymbol{j}} \end{bmatrix}. \tag{35}$$

Finally, the coordinate of the follower's desired formation position $r_{d,F}$ can be obtained by

$$\begin{aligned} \boldsymbol{r}_{d,F} &= \boldsymbol{r}_L + \boldsymbol{r}_{L2F,I} \\ \Rightarrow \{\boldsymbol{r}_{d,F}\}_{\boldsymbol{I}} &= \{\boldsymbol{r}_L\}_{\boldsymbol{I}} + \begin{bmatrix} \rho_\parallel & \rho_\perp \end{bmatrix} \begin{bmatrix} \cos\theta_{VL} & \sin\theta_{VL} \\ -\sin\theta_{VL} & \cos\theta_{VL} \end{bmatrix}. \end{aligned} \tag{36}$$

Applying Equation (36) to every points of the leader's position in the predicted trajectory, the desired formation trajectory of each follower in the horizontal motion over the predicted time horizon can be obtained.

### 5.1.2. Generation of the Desired Formation Trajectory in the Vertical Motion

In this study, followers are designated to maintain the same altitude as the leader throughout the flight. Thus, the leader's predicted trajectory of the altitude computed by the MPC algorithm directly becomes the followers' desired formation trajectories. The leader's vertical motion is determined by the desired altitude and/or climb rate, which can be classified into four cases as shown in Table 3.

**Table 3.** Cases of leader's vertical motion with desired altitude and climb rate.

| Case | Desired Altitude | Climb Rate | Vertical Motion |
|------|------------------|------------|-----------------|
| I | Not Given | Not Given | Maintain current altitude. |
| II | Given | Not Given | Climb to desired altitude. |
| III | Not Given | Given | Continuous climbing with given climb rate. |
| IV | Given | Given | Climb to desired altitude with given climb rate. |

In the vertical motion, the first two states, $\Delta z$ and $\Delta w$ in the state-space model (13) are used to construct a new second-order state-space model. The new second-order state-space model describes the leader's vertical motion and can be rewritten as follows:

$$\begin{bmatrix} \Delta\dot{z} \\ \Delta\dot{w} \end{bmatrix} = \begin{bmatrix} 0 & 1 \\ 0 & 0 \end{bmatrix} \begin{bmatrix} \Delta z \\ \Delta w \end{bmatrix} + \begin{bmatrix} 0 \\ \frac{-1}{m} \end{bmatrix} \Delta T_{total}. \tag{37}$$

$\Delta z$ is positive as the quadrotor moving downward. However, it is more intuitive for us to consider moving upward as positive. Hence, variable $\Delta h$ is defined to replace $\Delta z$ in the state-space model:

$$\Delta h = -\Delta z \quad \Rightarrow \quad \Delta \dot{h} = -\Delta w,$$

and therefore

$$\begin{bmatrix} \Delta \dot{h} \\ \Delta \dot{w} \end{bmatrix} = \begin{bmatrix} 0 & -1 \\ 0 & 0 \end{bmatrix} \begin{bmatrix} \Delta h \\ \Delta w \end{bmatrix} + \begin{bmatrix} 0 \\ \frac{-1}{m} \end{bmatrix} \Delta T_{total}. \tag{38}$$

With the given desired altitude and/or climb rate, the vertical reference trajectory of the leader for the MPC can be obtained. The MPC computes the control input and the predicted trajectory according to the reference trajectory and the state-space model (38). The following illustrate how the reference trajectory for the MPC is generated in each case in Table (3):

- Case I: maintain the current altitude.

  In this case, the leader is designated to maintain the altitude. Thus, the reference trajectories of $\Delta h$ and $\Delta w$ over the predicted time horizon are equal to zero.

- Case II: climb to the desired altitude.

  If the desired altitude is given but not the climb rate, the leader is designated to climb to the desired altitude with the default climb rate. The reference for $\Delta w$ is the negative default climb rate and the reference trajectory of $\Delta h$ is computed by

$$\boldsymbol{d}_{Alt}(k) = \overline{L}_p + RC_{default} \times k \times \Delta h_s, \tag{39}$$

  where $\boldsymbol{d}_{Alt}(k)$ denotes the reference altitude at time step $k$, which is equal to the leader's current altitude $\overline{L}_p$ plus the default climb rate $RC_{default}$ times the time step $k$ and the interval of the time step $\Delta h_s$. Time step $k$ is defined over the predicted time horizon considered by the MPC with the interval of $\Delta h_s$.

  A deceleration action is taken as the leader is approaching the desired altitude. If the distance between the leader's current position and the desired altitude is smaller than the altitude that can increase/decrease according to the climb rate within one second, then the reference of the altitude is set to the desired altitude and the reference of the climb rate will be set to zero.

- Case III: continuous climbing with the given climb rate.

  In this case, the reference trajectory of the altitude is generated by Equation (39) as well, but $RC_{default}$ is replaced by the given climb rate. No deceleration action is taken.

- Case IV: climb to desired altitude with the given rate.

  In this case, the reference trajectory of the altitude is generated by Equation (39) as well, but $RC_{default}$ is replaced by the given climb rate. This is the same if the statement in Case II is taken into consideration as the leader is approaching the desired altitude.

### 5.2. Formation Strategy: Follower

The follower's formation strategies in horizontal and vertical motions are illustrated in Figure 5. In follower's formation strategy, a follower is expected to receive two sets of data. The first data set includes the desired formation trajectories of itself and the followers that have a direct connection to it in the network structure. The second data set includes the states of the connected followers. How the blocks manipulate the received data sets and generate the control input is discussed in the following sections.

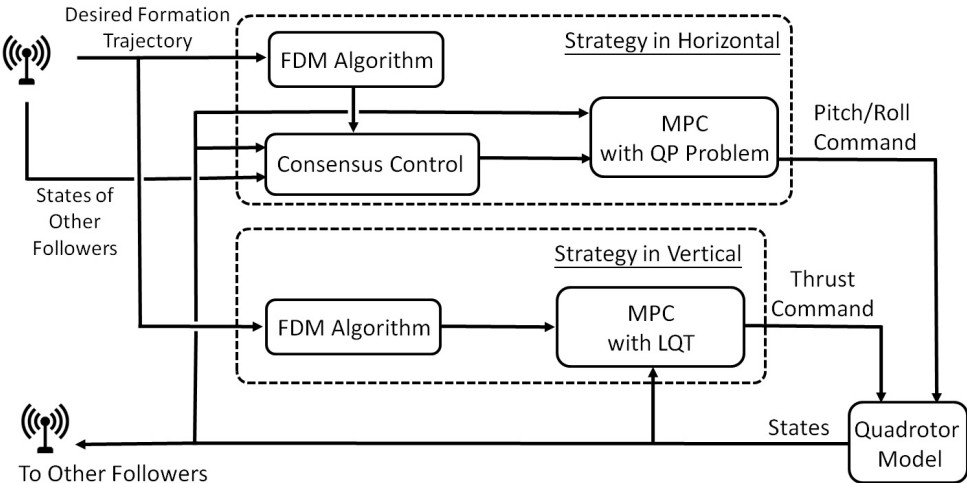

**Figure 5.** Follower's control strategy.

### 5.2.1. Formation Strategy in the Horizontal Motion

In the horizontal motion, the formation strategy is carried out by the consensus control and MPC. As mentioned in Section 2, the desired formation trajectories possess only the information of the desired position for the followers over the time horizon. Therefore, an algorithm of the finite difference method, the block FDM Algorithm in Figure 5, is added to the formation strategy in the horizontal motion to generate the required information for the consensus control.

Consider point $\zeta_i$ on given function $f(\zeta)$, the first to third derivative of $\zeta_i$ given by the finite difference method are shown as follows:

$$
\begin{aligned}
f'(\zeta_i) &= \frac{-f(\zeta_{i+2}) + 8f(\zeta_{i+1}) - 8f(\zeta_{i-1}) + f(\zeta_{i-2})}{12h}, \\
f''(\zeta_i) &= \frac{-f(\zeta_{i+2}) + 16f(\zeta_{i+1}) - 30f(\zeta_i) + 16f(\zeta_{i-1}) - f(\zeta_{i-2})}{12h^2}, \\
f'''(\zeta_i) &= \frac{-f(\zeta_{i+3}) + 8f(\zeta_{i+2}) - 13f(\zeta_{i+1}) + 13f(\zeta_{i-1}) - 8f(\zeta_{i-2}) + f(\zeta_{i-3})}{8h^3},
\end{aligned}
\tag{40}
$$

where $h$ is the time interval between each point. As shown in Equation (40), computing the third derivative of $f(\zeta)$ at $\zeta_i$ requires the points three steps prior to, and three steps after, $\zeta_i$. This would be a problem when dealing with the first three points and the last three points of the desired formation trajectory, since we don't have knowledge of the points prior to the first point and after the last point of the desired formation trajectory. However, these points can be acquired by applying linear extrapolation backward in time to the first point and forward in time to the last point of the desired formation trajectory. Once the complete desired trajectory is generated, consensus control engages. The consensus control algorithm computes the flight paths of the followers that track the corresponding desired formation trajectories using the control input given in Equation (15). Here, we rearrange the states at time step $k$ of the i-th follower as follows:

$$
s_i^{(1)}(k) = \begin{bmatrix} \Delta x(k) \\ \Delta y(k) \end{bmatrix}, \quad
s_i^{(2)}(k) = \begin{bmatrix} \Delta u(k) \\ \Delta v(k) \end{bmatrix}, \quad
s_i^{(3)}(k) = \begin{bmatrix} \Delta \theta(k) \\ \Delta \phi(k) \end{bmatrix}, \quad
s_i^{(4)}(k) = \begin{bmatrix} \Delta q(k) \\ \Delta p(k) \end{bmatrix}. \tag{41}
$$

The corresponding desired formation trajectory of each modified state $s_i^{(ks)}(k)$ ($ks = 1, 2, 3, 4$) at time step $k$ is defined as $d_i^{(1)}(k)$, $d_i^{(2)}(k)$, $d_i^{(3)}(k)$ and $d_i^{(4)}(k)$. The error between the modified state and the corresponding desired formation trajectory at time step $k$ can be written as:

$$
\hat{s}_i^{(ks)}(k) = s_i^{(ks)}(k) - d_i^{(ks)}(k), \quad ks = 1, 2, 3, 4. \tag{42}
$$

The consensus control input at time step $k$ can be computed by using the modified states and the control gains $\beta_{ks}$ ($ks = 1, 2, 3, 4$). The control input will result in a two-dimensional column matrix. The first and second elements of the control input are the pitch and roll control commands, respectively. The next step is to apply the consensus control input to the discrete state-space model of (13). Repeating the steps above until the flight path over the predicted time horizon is obtained.

After the flight path of the follower under consideration is generated, it then serves as the reference trajectory for the MPC. The MPC obtains the optimal control sequence using Equation (29). First, rearrange the initial condition and the reference trajectory in the form of $\bar{x}$ described in Equation (27). With the discrete state-space model and the selected weighting matrices, the predicted trajectory and the control input sequence can be obtained by Equations (23) and (29). Then, the first control input of the optimal control sequence becomes the formation control input for the follower.

### 5.2.2. Formation Strategy in the Vertical Motion

The formation strategy in the vertical motion is simpler than the one in the horizontal motion. The state-space model that is adopted in the vertical motion is describe by Equation (38). As mentioned in Section 2, the desired formation trajectory generated by the leader possesses only the information of the altitude, namely $\Delta h$. Therefore, an algorithm of the finite-difference method, block *FDM Algorithm* in Figure 5 is added to the follower's vertical formation strategy to generate the required information for the LQT.

With the complete desired formation trajectory and the selected weighting matrices, the LQT computes the optimal control sequence using Equation (19). The formation control input in the vertical motion for the follower is the first input of the optimal control sequence.

## 6. Simulation Preparation

This section introduces the configuration of the systems that are used in the simulation under the Matlab/Simulink environment, as well as the necessary setup for the simulation. A transformation of variables is carried out in Section 6.1 for coding purpose. In Section 6.2, the configuration of the simulation systems under the Matlab/Simulink environment are introduced block by block and leader to follower. To mention, one leader, three followers, and the network structure in Figure 1b are considered in the simulation.

### 6.1. Transformation of the Variable of the Discrete State-Space Model

The state-space model coded in the formation strategies is the discretized state-space model with a sampling time of 0.1 s. That is, the controls for the leader and the followers are expected to be updated every 0.1 s. For the convenience of coding the control algorithm in the horizontal motion, a transformation of variables is applied to the horizontal state-space model (13). The states and the controls in the horizontal state-space model (13) are arranged as $x_h$ and $U_h$ in Equation (43), respectively. The transformation of states $T_s$ and the transformation of controls $T_u$ are considered and shown as follows:

$$
x_h = \begin{bmatrix} \Delta x \\ \Delta y \\ \Delta u \\ \Delta v \\ \Delta \theta \\ \Delta \phi \\ \Delta q \\ \Delta p \end{bmatrix} = \begin{bmatrix} 1 & 0 & 0 & 0 & 0 & 0 & 0 & 0 \\ 0 & 1 & 0 & 0 & 0 & 0 & 0 & 0 \\ 0 & 0 & 1 & 0 & 0 & 0 & 0 & 0 \\ 0 & 0 & 0 & 1 & 0 & 0 & 0 & 0 \\ 0 & 0 & 0 & 0 & \frac{-1}{g} & 0 & 0 & 0 \\ 0 & 0 & 0 & 0 & 0 & \frac{1}{g} & 0 & 0 \\ 0 & 0 & 0 & 0 & 0 & 0 & \frac{-1}{g} & 0 \\ 0 & 0 & 0 & 0 & 0 & 0 & 0 & \frac{1}{g} \end{bmatrix} \begin{bmatrix} \Delta x \\ \Delta y \\ \Delta u \\ \Delta v \\ \Delta \widetilde{\theta} \\ \Delta \widetilde{\phi} \\ \Delta \widetilde{q} \\ \Delta \widetilde{p} \end{bmatrix} = T_s \widetilde{x}_h,
$$

$$
U_h = \begin{bmatrix} \Delta M_\theta \\ \Delta M_\phi \end{bmatrix} = \begin{bmatrix} \frac{-I_{yy}}{g} & 0 \\ 0 & \frac{I_{xx}}{g} \end{bmatrix} \begin{bmatrix} \Delta \widetilde{M}_\theta \\ \Delta \widetilde{M}_\phi \end{bmatrix} = T_u \widetilde{U}_h.
$$

(43)

Applying the above transformation to the states and the controls in Equation (13), the normalized horizontal state-space model considering the modified state $\widetilde{x}_h$ and the modified control $\widetilde{U}_h$ can be obtained. The normalized system matrix $\widetilde{A}_h$ and the control matrix $\widetilde{B}_h$ are shown as follows:

$$
\widetilde{A}_h = \begin{bmatrix} 0 & 0 & 1 & 0 & 0 & 0 & 0 & 0 \\ 0 & 0 & 0 & 1 & 0 & 0 & 0 & 0 \\ 0 & 0 & 0 & 0 & 1 & 0 & 0 & 0 \\ 0 & 0 & 0 & 0 & 0 & 1 & 0 & 0 \\ 0 & 0 & 0 & 0 & 0 & 0 & 1 & 0 \\ 0 & 0 & 0 & 0 & 0 & 0 & 0 & 1 \\ 0 & 0 & 0 & 0 & 0 & 0 & 0 & 0 \\ 0 & 0 & 0 & 0 & 0 & 0 & 0 & 0 \end{bmatrix}, \quad \widetilde{B}_h = \begin{bmatrix} 0 & 0 \\ 0 & 0 \\ 0 & 0 \\ 0 & 0 \\ 0 & 0 \\ 0 & 0 \\ 1 & 0 \\ 0 & 1 \end{bmatrix}.
$$

(44)

The discrete state-space model that is adopted by the formation strategies in the horizontal motion is obtained by applying the Matlab function *c2d* to the normalized continuous state-space model (44).

### 6.2. Simulink Environment Setup for the Formation Simulation

The basic configuration of the simulated quadrotor system under the Matlab/Simulink environment is shown in Figure 6. The system includes a *Control System* module, a *Gravity Influence* module, and a six degrees of freedom (6DOF) model module (*Quad 6DOF Model*). Both leader and followers adopt the same configuration with the difference in the Control System module and the input/output ports.

The *Gravity Influence* module computes the distribution of the quadrotor's weight on the axes of frame **B**. The 6DOF rigid body EOMs solver inside the *Quad 6DOF Model* module is shown in Figure 7. Block *6DOF (Euler Angles)* solves the nonlinear 6DOF EOMs of the rigid body with the given inputs with respect to frame **B**. The block is provided by the Simulink/Aerospace Blockset toolbox [17].

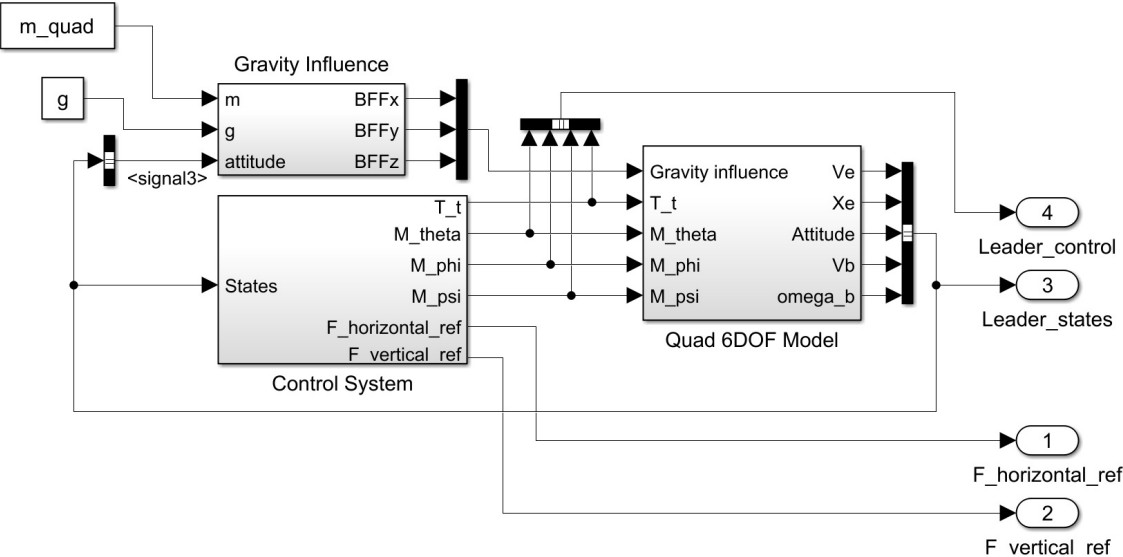

**Figure 6.** Simulated quadrotor system: leader.

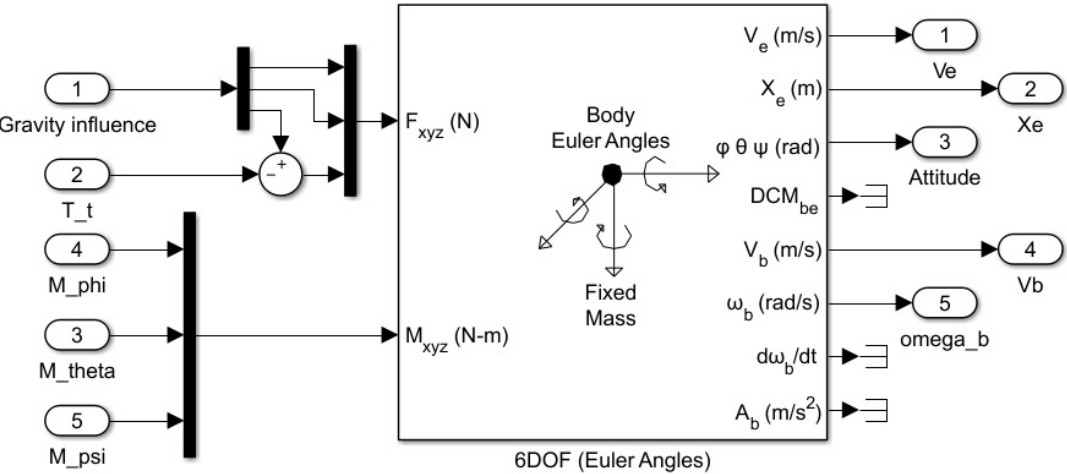

**Figure 7.** Simulink/Aerospace Blockset 6DOF block.

The *Control System* module basically includes a yaw control submodule, pitch/roll control submodule, and an altitude control submodule. Figure 8 shows the configuration of the leader's *Control System* module. The follower's *Control System* module differs from the leader's *Control System* module in the pitch/roll control submodule. The leader's and follower's pitch/roll control submodule will be introduced later in the section. The yaw control submodule controls the yaw motion of the quadrotor based on the dual-loop proportional plus integral (PI) control. The pitch/roll and altitude controls carry out the formation control strategies depending on the role that it is a leader or a follower. Likewise, the input/output ports differ depending on the role of the quadrotor.

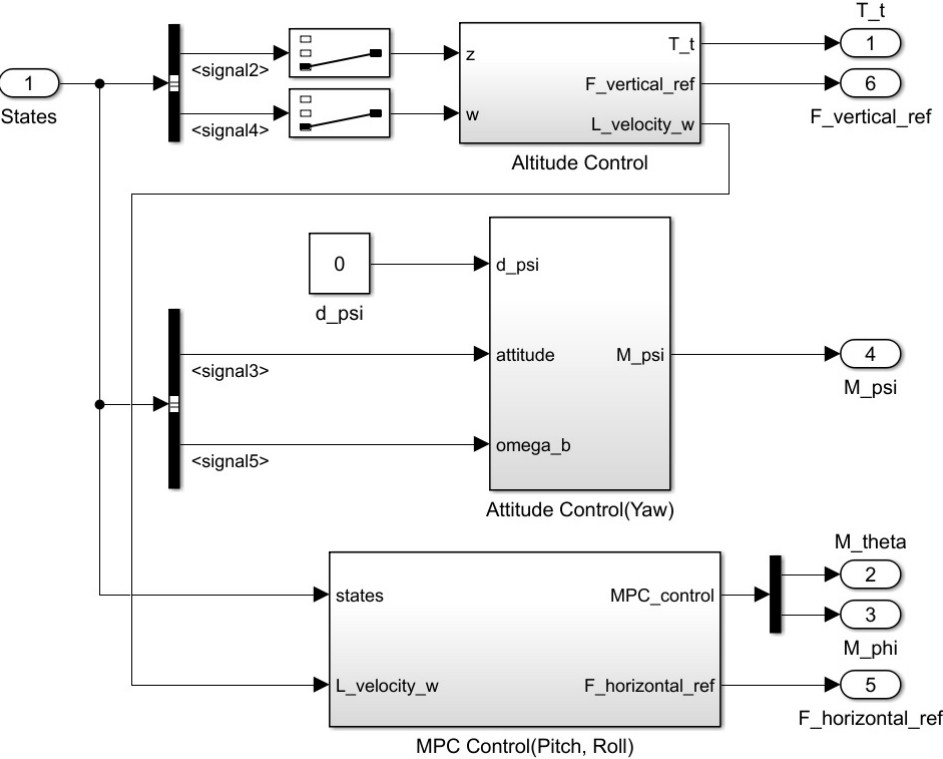

**Figure 8.** Configuration of the control system.

Figure 9 shows the configuration of the *Altitude Control* submodule of the leader and the follower. The *Altitude Control* submodule carries out the vertical formation strategies described in Section 5.1.2 for the leader and in Section 5.2.2 for the follower. In the leader's *Altitude Control* submodule, the function *Track Generator* generates the reference trajectory *Lv_ref* for the LQT using the method provided in Section 5.1.2 depending on the vertical state of the leader, desired altitude, and climb rate. The vertical state of the leader, labeled as *vN*, is defined by Equation (38). The desired altitude and the climb rate, labeled as *required_alt* and *required_RC*, are given by the user. Function *LQT* tracks reference trajectory *Lv_ref* by solving Equation (19) with the given weighting matrices, vertical state *vN*, and vertical state-space model (38). Function *LQT* outputs thrust command *dT*, the follower's reference trajectory *F_vertical_ref*, and leader's velocity *L_velocity_w* on $\hat{b}_3$ of frame **B**. In the follower's *Altitude Control* submodule, function *FDM Algorithm* computes the complete reference trajectory *Fv_ref* consisting of the references for states $\Delta h$ and $\Delta w$ in Equation (38) using the finite-difference method (40). Then, function *LQT* solves the equations in (19) depending on the given weighting matrices, the vertical state of follower *vN*, and vertical state-space model (38). After the computation, function *LQT* outputs the thrust command *dT*. It is worth mentioning that the thrust command *dT* in both the leader's and follower's control submodules is the thrust deviation from the thrust at the equilibrium point $T_{total,0}$. Therefore, actual thrust command *T_t* in Figure 6 that is input into the 6DOF model equals the thrust deviation *dT* plus the equilibrium thrust $T_{total,0}$. Leader's velocity *L_velocity_w* is used to compute the leader's velocity in frame **I** in the control strategy in horizontal motion.

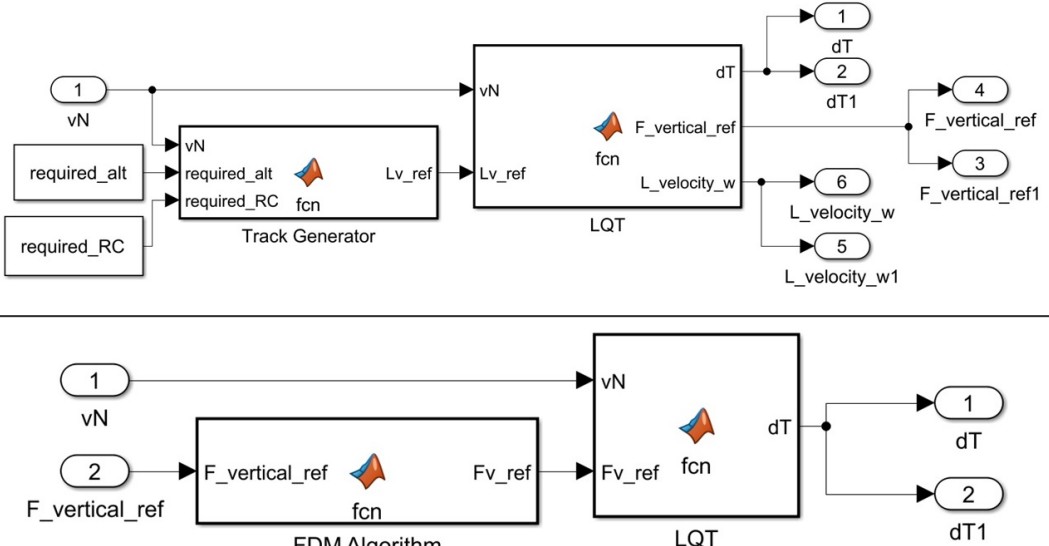

**Figure 9.** Configuration of altitude control of leader (**top**) and follower (**bottom**).

Figures 10 and 11 show the pitch/roll control submodule for the leader and the follower, respectively. The leader's pitch/roll control submodule, *MPC Control(Pitch, Roll)* in Figure 8, includes the waypoints datasets (orange dashed square), the waypoints update rules (orange square), the block *Preset Constant* which includes the formation pattern and the parameters for the waypoint update rules, the MPC algorithm *Leader MPC Algorithm*, and the function *Follower ref. Trajectory Generator* that generates the followers' reference trajectories as shown in Figure 10. The waypoints datasets in the orange dashed square includes the coordinate of the waypoints in frame *I*, named as *L_wp*, and the number of the waypoints, named as *wp_length*. The waypoint update rules update the waypoint if the distance between the leader and the waypoint, denoted by *R2wp*, is smaller than $d_2$ and hold at the last waypoint if *R2wp* is smaller than $d_2$. Parameter $d_1$ is used under the circumstances that if the user designates the followers to hold a specific position in which the last waypoint is reached. Function *Leader MPC Algorithm* solves the optimal tracking control input *M* by using Equation (29) with the given weighting matrices, the horizontal state of the leader, and the horizontal state-space model. The predicted trajectory for the leader is computed by Equation (23). Note that the horizontal state and the state-space model that are adopted in the *Leader MPC Algorithm* are the modified state $\tilde{x}_h$ and the normalized state-space model in Equation (44). In addition, the control output by the *Leader MPC Algorithm* is the modified control in Equation (43). Function *Follower ref. Trajectory Generator* computes the reference trajectories for the followers based on the method in Section 5.1.1 using the leader's predicted trajectory and the formation pattern. The leader's vertical velocity *L_velocity_w* is input into the function to compute the leader's velocity in frame *I* using Equation (8).

The control submodule shown in Figure 11 is the control submodule of Follower #1. The follower's control submodule first computes the complete desired formation trajectories, including the information of the velocity, attitude, and angular velocity, using the finite difference method coded in function *FDM Algorithm*. Then, the function *Consensus Algorithm* computes all the followers' flight paths over the predicted time horizon based on the consensus control (15), modified state $\tilde{x}_h$ (43), and the normalized state-space model (44). The signal, labeled as *rN*, is the modified state of the follower under consideration and the signal, labeled as *rNF_all*, contains all the modified states of other followers connected to it in the network structure. In the case shown in Figure 11, *rN* is the modified state of Follower #1 and *rNF_all* contains the modified states of Follower #2 and Follower #3. After the flight paths are computed, the function *MPC algorithm* engages. The *MPC algorithm* computes the optimal tracking control by using Equadtion (29) based on the modified state (43), the normalized state-space model (44) of the quadrotor, and the corresponding flight path.

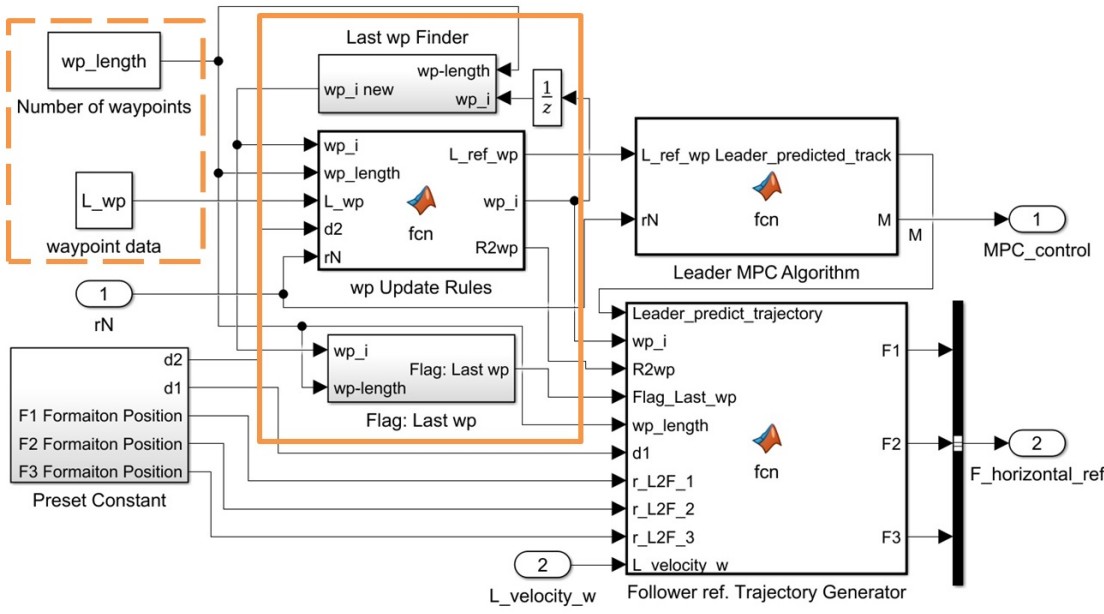

**Figure 10.** Configuration of leader's horizontal motion control.

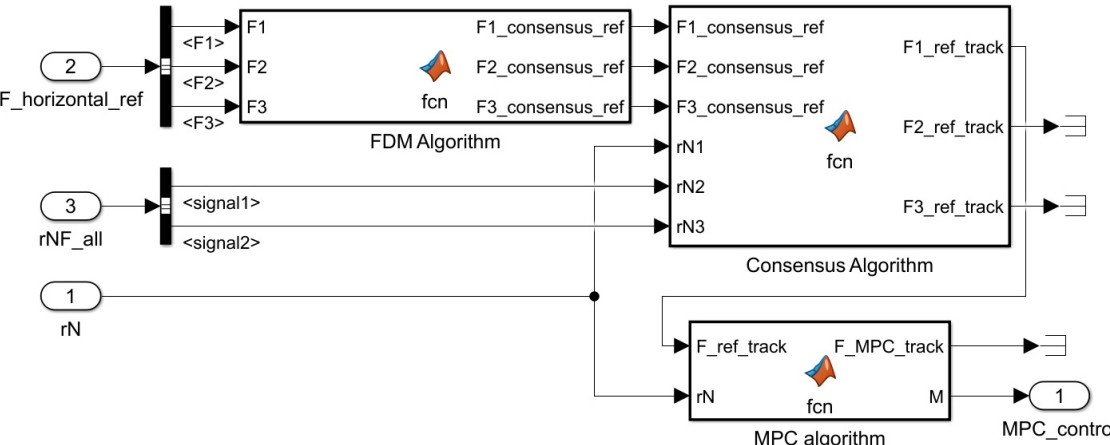

**Figure 11.** Configuration of follower's horizontal motion control.

## 7. Simulation Results

The simulation of a comprehensive maneuver is carried out to examine the performance of the proposed formation strategies. The parameters of the quadrotor, the gains of the consensus control, and the weighting matrices that are used in the simulation are given in Section 7.1. The simulation results of the comprehensive maneuver are shown in Section 7.2.

### 7.1. Quadrotor Parameter, Control Gain, and Weighting Matrices

This section summarizes the parameters of the quadrotor (Table 4), weighting matrices, and consensus control gains that are used in the simulation.

**Table 4.** Parameters of the quadrotor.

| Parameter | Symbol | Value | Unit |
|---|---|---|---|
| Mass of the Quadrotor | m | 1.5 | $kg$ |
| Acceleration of Gravity | g | 9.807 | $m/s^2$ |
| Quadrotor Arm Length | L | 0.225 | $m$ |
|  | $I_{xx}$ | 0.022 | $kg \cdot m^2$ |
| Moment of Inertia | $I_{yy}$ | 0.022 | $kg \cdot m^2$ |
|  | $I_{zz}$ | 0.0018 | $kg \cdot m^2$ |

Note: the above parameters were acquired from the quadrotor in the lab.

In the simulation, we assume that all the states in both horizontal and vertical motions are obtainable. That is, output matrix $C$ in the state-space model for both motions for leader and followers is an identity matrix. The weighting matrices for the LQT and QP problem for the leader are given as follows:

- Weighting matrices $P$, $Q$, and $R$ of the unconstrained QP problem coded in simulink function *Leader MPC Algorithm* in Figure 10 for the leader's horizontal motion control are given as follows:

$$
\begin{aligned}
Q &= I_8, \\
R &= 0.5 I_2, \\
P &= \begin{bmatrix}
29.8176 & 0 & 37.9636 & 0 & 25.3123 & 0 & 7.1204 & 0 \\
0 & 29.8176 & 0 & 37.9636 & 0 & 25.3123 & 0 & 7.1204 \\
37.9636 & 0 & 86.4879 & 0 & 67.0493 & 0 & 20.8541 & 0 \\
0 & 37.9636 & 0 & 86.4879 & 0 & 67.0493 & 0 & 20.8541 \\
25.3123 & 0 & 67.0493 & 0 & 75.6689 & 0 & 26.9672 & 0 \\
0 & 25.3123 & 0 & 67.0493 & 0 & 75.6689 & 0 & 26.9672 \\
7.1204 & 0 & 20.8541 & 0 & 26.9672 & 0 & 18.4432 & 0 \\
0 & 7.1204 & 0 & 20.8541 & 0 & 26.9672 & 0 & 18.4432
\end{bmatrix}
\end{aligned}
\tag{45}
$$

- Weighting matrices $P$, $Q$, and $R$ for the LQT coded in the simulink function *LQT* in Figure 9 for the leader's altitude control are given as follows:

$$
P = Q = \begin{bmatrix} 3 & 0 \\ 0 & 10 \end{bmatrix}, \quad R = 0.5.
\tag{46}
$$

As for the follower, the control gains for the consensus control in the horizontal motion and the weighting matrices for the LQT in vertical motion and the unconstrained QP problem in horizontal motion are given as follows:

- Consensus control gain $\beta_{ks}$ ($ks = 1, 2, 3, 4$) in the horizontal motion considering the states in (41) and the consensus control (15) coded in the *Consensus Algorithm* in Figure 11 are given as follows:

$$
(\beta_1, \beta_2, \beta_3, \beta_4) = (13, 30, 60, 5).
\tag{47}
$$

- Weighting matrices $P$, $Q$, and $R$ of the unconstrained QP problem coded in simulink function *MPC Algorithm* in Figure 11 for the follower's horizontal motion control are given as follows:

$$
\begin{aligned}
Q &= I_8, \\
R &= 0.3I_2, \\
P &= \begin{bmatrix}
28.8987 & 0 & 35.3119 & 0 & 22.1493 & 0 & 5.5294 & 0 \\
0 & 28.8987 & 0 & 35.3119 & 0 & 22.1493 & 0 & 5.5294 \\
35.3119 & 0 & 78.6321 & 0 & 57.3341 & 0 & 15.6844 & 0 \\
0 & 35.3119 & 0 & 78.6321 & 0 & 57.3341 & 0 & 15.6844 \\
22.1493 & 0 & 57.3114 & 0 & 62.9640 & 0 & 19.4694 & 0 \\
0 & 22.1493 & 0 & 57.3114 & 0 & 62.9640 & 0 & 19.4694 \\
5.5294 & 0 & 15.6844 & 0 & 19.4694 & 0 & 12.6822 & 0 \\
0 & 5.5294 & 0 & 15.6844 & 0 & 19.4694 & 0 & 12.6822
\end{bmatrix}
\end{aligned}
\tag{48}
$$

- Weighting matrices *P*, *Q*, and *R* for the LQT coded in simulink function *LQT* in Figure 9 for the follower's altitude control are given as follows:

$$
Q = P = \begin{bmatrix} 3 & 0 \\ 0 & 10 \end{bmatrix}, \quad R = 0.5.
\tag{49}
$$

The selection of weighting matrix *P* for the unconstrained QP problem utilizes the methodology provided by Reference [18]. The method ensures the MPC objective function has the same quadratic cost as the infinite horizon quadratic cost used by LQR.

### 7.2. Simulation of the Comprehensive Maneuver

A simulation of the comprehensive maneuver that was carried out is outlined in this section along with the simulation results. The objective of the simulation was to examine whether the followers can carry out the formation flight, including maintaining the given formation pattern and tracking the desired altitude, with the proposed formation strategies while the leader performs a series of maneuvers.

Assume that two quests, mapping the landscape and taking photos of a target building's exterior, are assigned to the leader. The predefined waypoints are illustrated in Figure 12. The first quest, mapping the landscape, is carried out through Waypoint #1 to Waypoint #9. An imaginary terrain is placed at waypoint #6 so that the leader has to change its altitude in order to cross the terrain. The leader is designated to climb to 20 meters with respect to the zeros reference altitude at the climb rate of 4 m/s at Waypoint #5 and descend to the zeros reference altitude at the climb rate of –4 m/s at Waypoint #7. The second quest, taking photos of a target building's exterior, is carried through Waypoint #15 to Waypoint #32. The leader follows a circle trajectory in the *xy* plane and climbs at the climb rate of 0.45 m/s at the same time. The climbing continues until the leader reaches 60 meters with respect to the zeros reference altitude. After all the quests are completed, the leader heads to its holding point at Waypoint #47 by following the Waypoint #33 to Waypoint #47. The leader will start to descend to zero reference altitude at Waypoint #38 at the climb rate of –2 m/s.

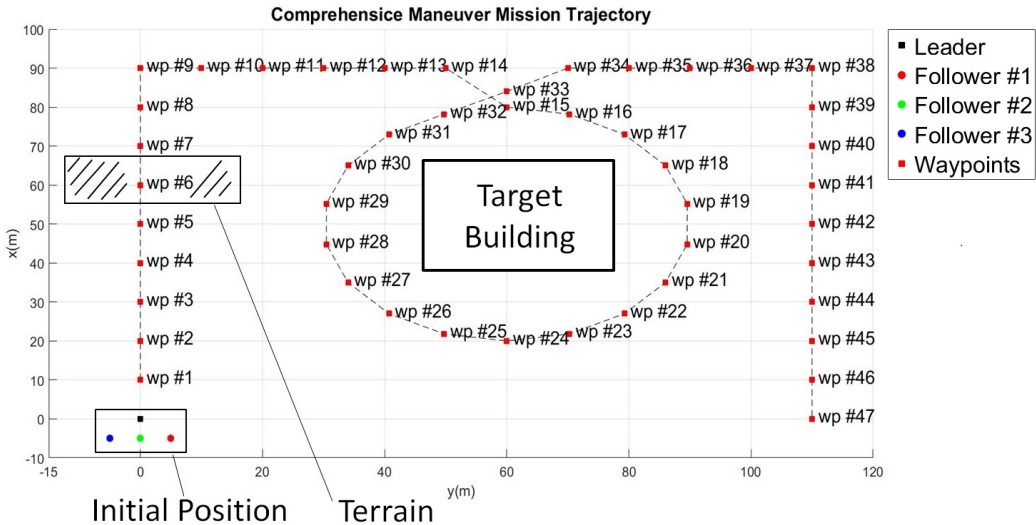

**Figure 12.** Comprehensive maneuver mission trajectory.

The triangle formation pattern is chosen and the simulation results are shown in the following figures. The trajectories of the quadrotors throughout the maneuver are shown in Figure 13. Figure 14 shows the top view of the quadrotors with the formation pattern throughout the flight. In Figures 13 and 14, the *x*-axis points to the north, the *y*-axis points to the east, and the *z*-axis points upward, opposite to the $\hat{k}$ axis in frame *I*. Figure 15 shows the altitude and climb rate of the quadrotors. The attitude of the leader and followers are shown in Figures 16–18.

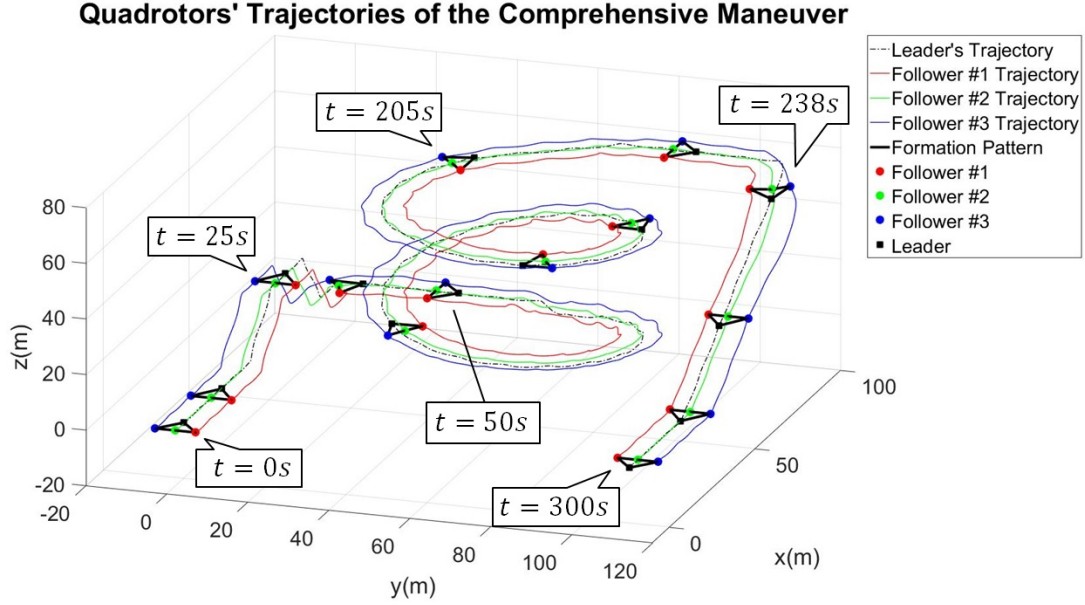

**Figure 13.** Quadrotor trajectories of comprehensive maneuvers.

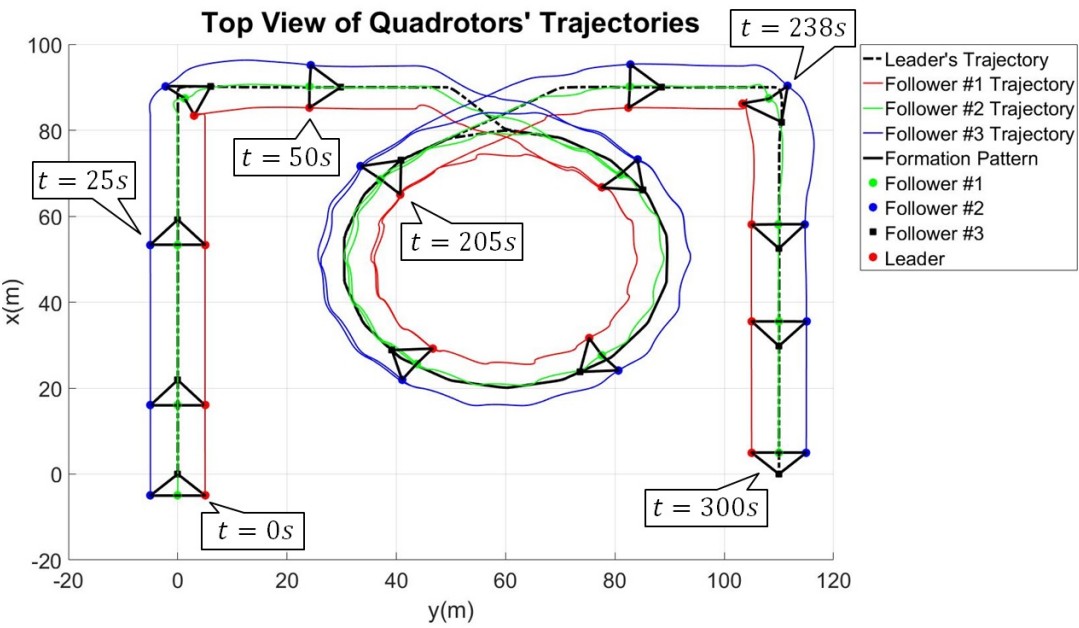

**Figure 14.** Top view of quadrotor trajectories throughout the comprehensive maneuver.

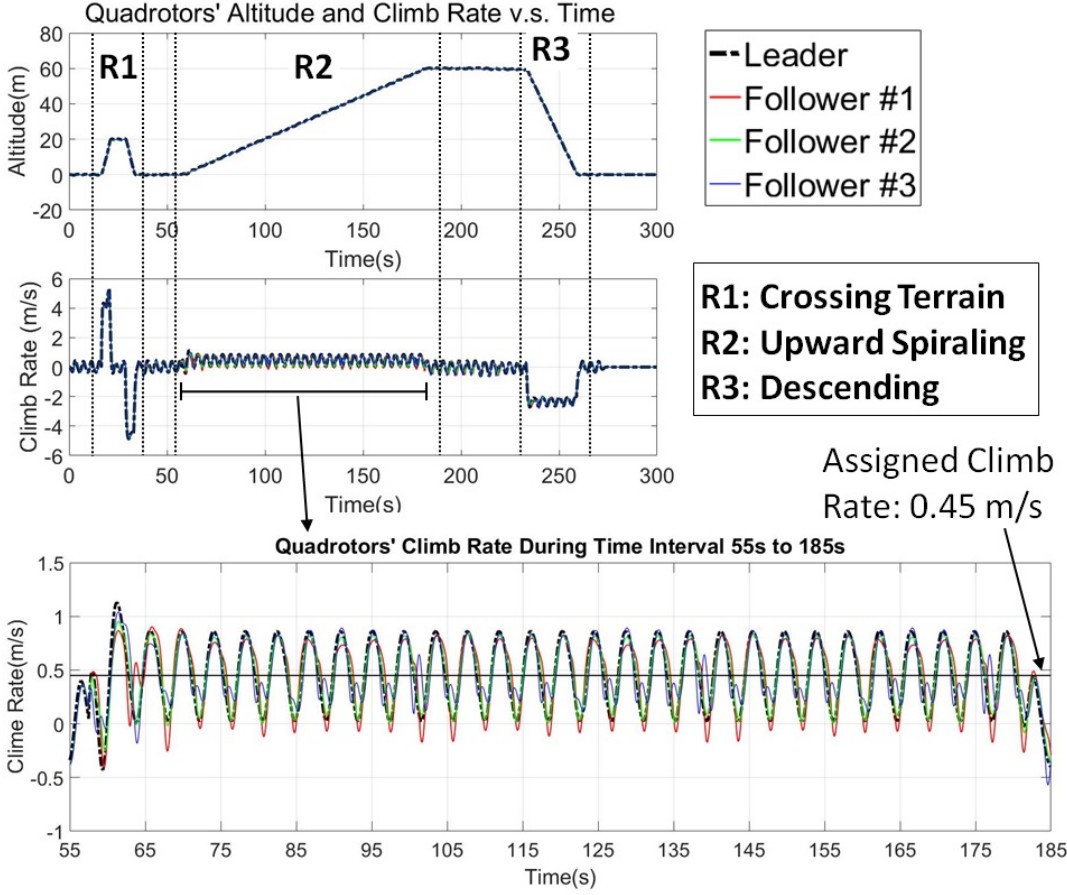

**Figure 15.** Quadrotor altitude and climb rate throughout the comprehensive maneuver.

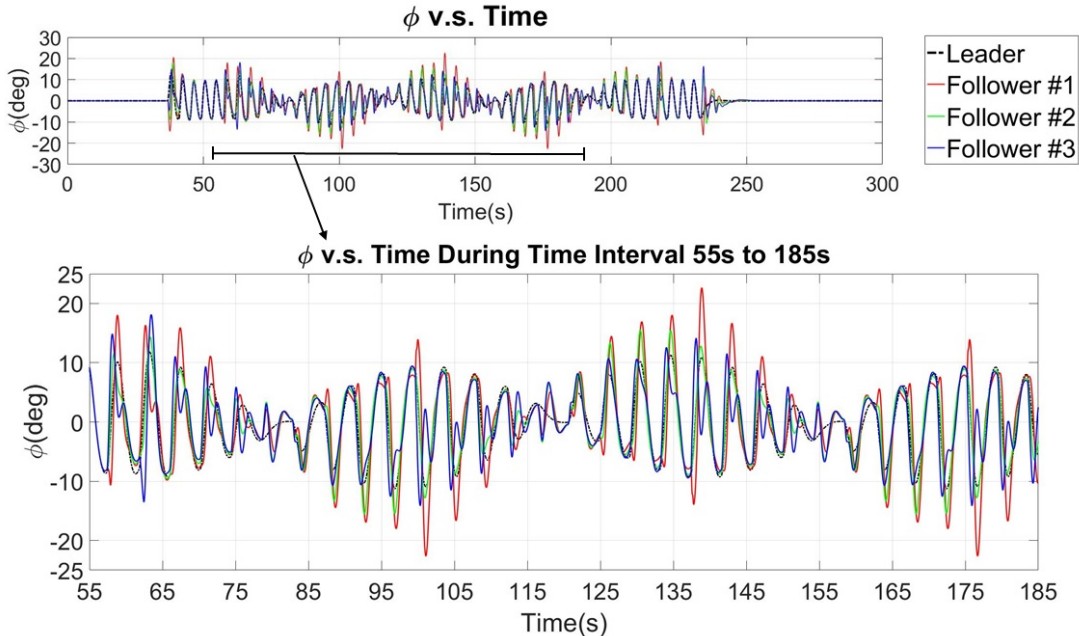

**Figure 16.** Quadrotor roll angle throughout the comprehensive maneuver.

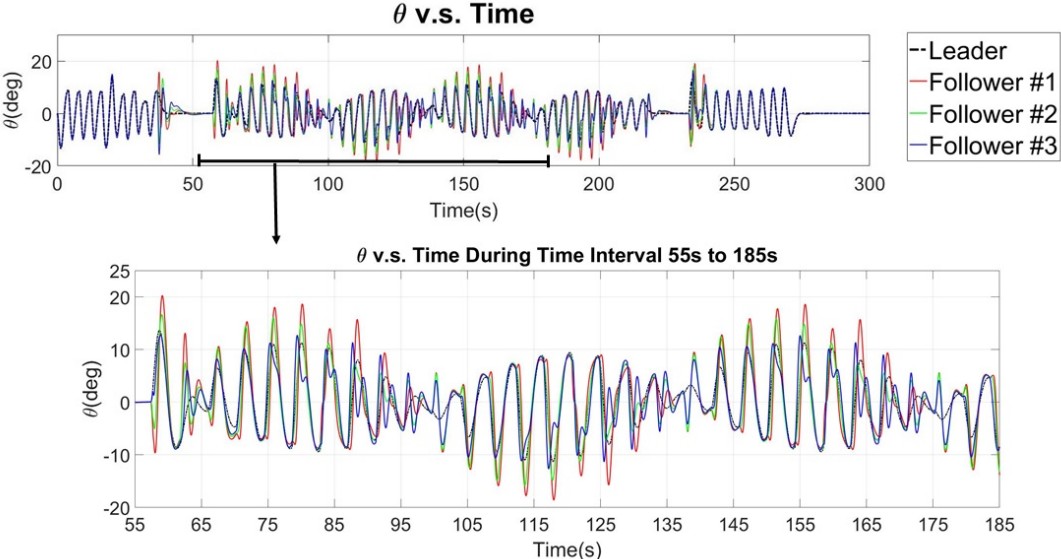

**Figure 17.** Quadrotor pitch angle throughout the comprehensive maneuver.

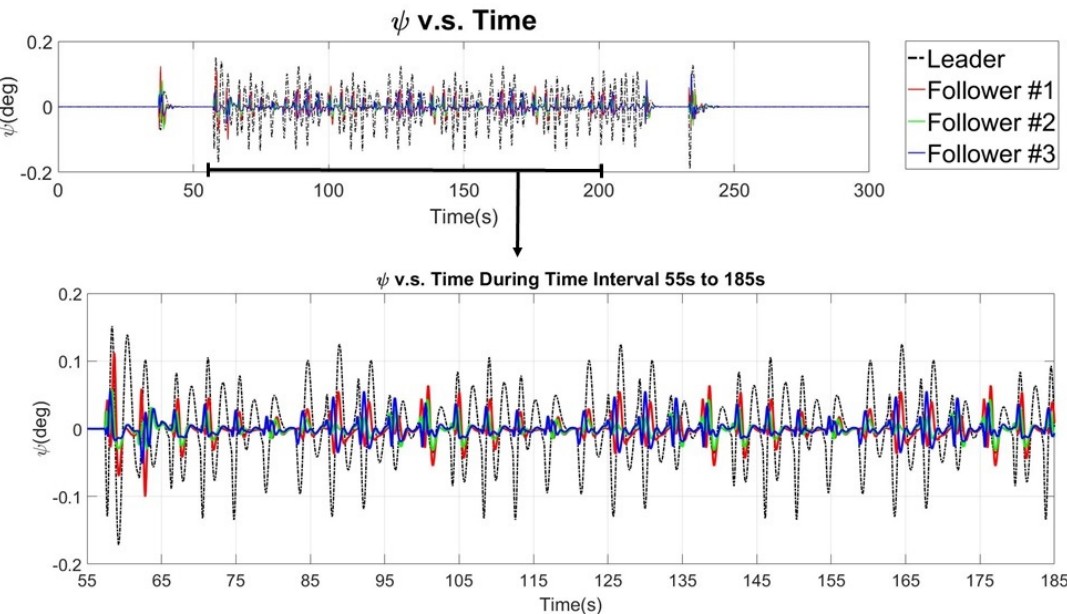

**Figure 18.** Quadrotor yaw angle throughout the comprehensive maneuver.

## 8. Discussion

In the simulation of the comprehensive maneuver, the leader performs a series of maneuvers while carrying out the assigned quests. The quadrotor trajectories in Figure 13 show that the followers carry out the formation flight well based on the proposed formation strategies. Furthermore, from the top view of the quadrotor trajectories in Figure 14, the designated formation pattern is maintained throughout the flight. Although the formation pattern slightly deforms during the turn, it recovers soon after the turn.

In the vertical motion, the leader climbs to the desired altitude at the designated climb rate in three stages, **R1**, **R2**, and **R3**, as shown in Figure 15. Moreover, the followers can keep up with the leader's vertical motion with the proposed formation strategy in Section 5.2.2. In Figure 15, oscillation in the altitude and climb rate can be observed throughout the flight. The oscillation can be seen more clearly in Figure 19, which is a zoom-in view of the vertical motion responses in Figure 15 during the time interval of 190 to 210 seconds. The oscillation is caused by the coupling effects between the horizontal and vertical motions in the 6DOF nonlinear EOMs. The quadrotor tilts while tracking the waypoints as shown in Figures 16 and 17, which disturbs the equilibrium condition and causes the altitude to change. Since the equilibrium condition is disturbed, the altitude control algorithm will engage to regain the equilibrium condition in the vertical motion and therefore result in the oscillating phenomenon.

In the horizontal motion, the responses of the pitch and roll angles in Figures 16 and 17 show that the angles remain in an acceptable range ($\pm25°$) throughout the flight. Furthermore, the yaw angle is well-controlled, close to zero, by the dual-loop PI control as shown in Figure 18. Additionally, the responses of the pitch and roll angles before and after the upward circling motion are shown in Figures 20 and 21, respectively. Before the quadrotors initiate and after the quadrotors finish the upward circling motion, they are simply performing a straight flight with a 90° turn. According to the results illustrated in Figures 20 and 21, the formation strategies in the horizontal motion are fully adequate to perform the formation in straight flight. Even during the turn, the pitch and roll angles stay in an acceptable range ($\pm25°$).

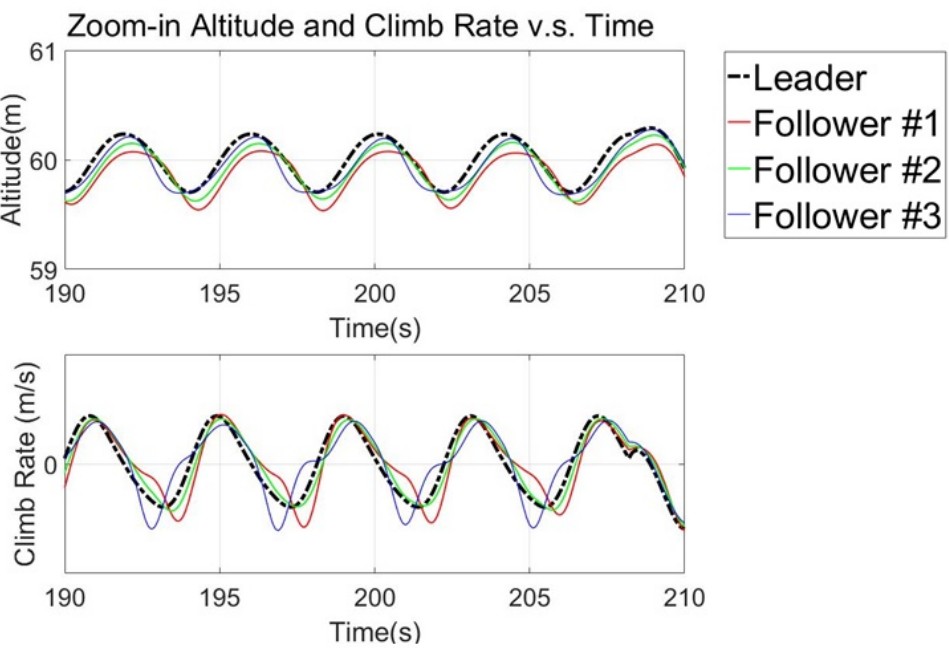

**Figure 19.** Zoom-in of the responses of altitude and climb rate during 190 to 210 seconds.

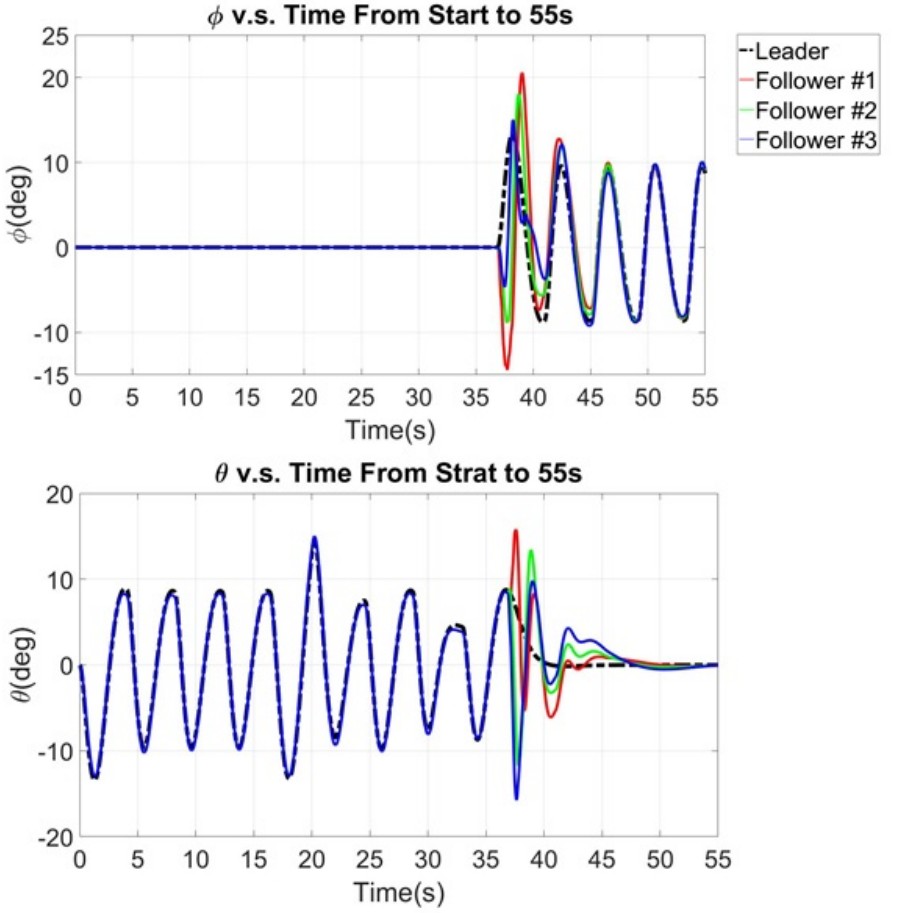

**Figure 20.** Responses of pitch and roll angles from start to 55 seconds.

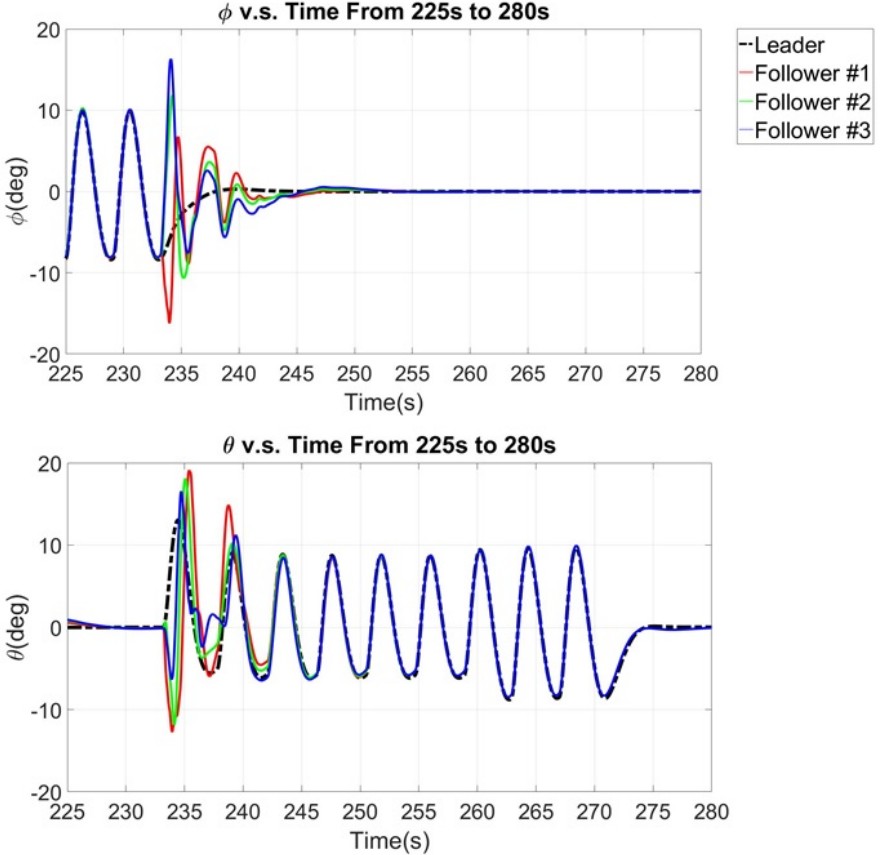

**Figure 21.** Responses of pitch and roll angles from 255 to 280 seconds.

A zoom-in figure of the responses of the attitude during the upward circling motion is shown in Figure 22. The figure zooms in the responses in the time interval of 80 to 100 seconds during the upward circling motion. In this 20-second interval, the quadrotors travel about 1/4 of a circle in an *xy* plane. During the upward circling motion, the leader has to change its direction frequently to stay on the circle track as well as the followers to stay in formation. The results in Figure 22 show that the pitch and roll angles are still within an acceptable range for leader and followers. The followers can keep up with leader's horizontal motion, or, in other words, to stay in formation during the upward circling motion with the proposed formation strategy in Section 5.2.1. The yaw angles of the leader and followers are controlled closed to zero with the dual-loop PI control in the upward circling motion as shown in Figures 18 and 22.

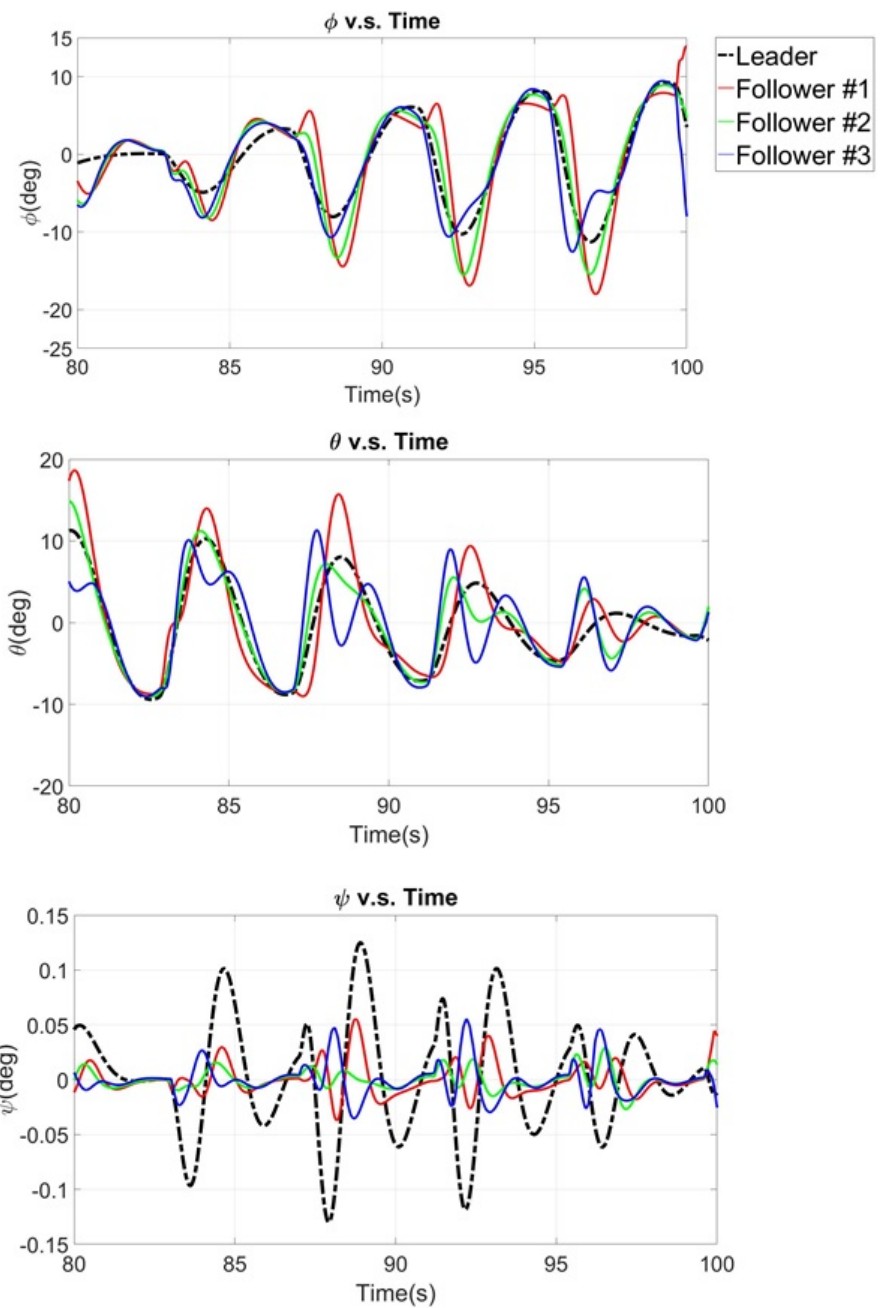

**Figure 22.** Zoom-in of altitude vs. time during upward circling motion.

## 9. Conclusions

In this study, the formation control strategies for quadrotors considering the leader–follower architecture are proposed in Section 5. The leader is designated to track the predefined waypoints in horizontal motion and the desired altitude/climb rate in vertical motion. Furthermore, the leader generates the followers' desired formation trajectories using the methods in Sections 5.1.1 and 5.1.2 as a guidance for the followers to perform the formation flight. The follower's formation strategy is proposed in Section 5.2. The consensus control and MPC with the QP problem are adopted in the horizontal formation control, and the MPC with the LQT is adopted in the vertical formation control. The formation control strategies are examined based on the 6DOF nonlinear simulation under the Matlab/Simulink environment and the simulation results are illustrated in Section 7. A discussion of the simulation results as well as the performance of the formation strategies are carried out in Section 8.

The results of the simulation of comprehensive maneuver in Section 7.2 show that the proposed formation strategies are feasible in straight flight, turning, and upward circling motion.

**Author Contributions:** J.-K.S. organized and supervised the main research of the study. C.-W.C. completed the design and simulation of the study under Shiau's supervision.

**Acknowledgments:** This research was supported by the Ministry of Science and Technology (MOST), Taiwan, under Grant MOST 107-2221-E-032-028.

**Conflicts of Interest:** The authors declare no conflict of interest. The founding sponsors had no role in the design of the study; in the collection, analyses, or interpretation of data; in the writing of the manuscript; and in the decision to publish the results.

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
