# Peer review of "Quadrotor Formation Strategies Based on Distributed Consensus and Model Predictive Controls"

_applsci, doi:10.3390/app8112246_

Round 1
Reviewer 1 Report
This research proposed a distributed consensus control and model predictive control (MPC) based control of quadrotors. The formation control strategies for leader and follower were also proposed. The authors concluded the formation control strategies proposed in the research area capable of carrying out complex formation but still satisfying the mission requirements with deeper study and research. The reviewer believes that the current version of the manuscript is not yet ready for publication; the authors are encouraged to consider the following comments and suggestion and revise the manuscript accordingly.
1. The authors should consider splitting the Introduction section into two sections, including an Introduction section and a Background (related work) section. The introduction section should focus on introducing the research objectives and the research questions that need to be addressed, while the Background section should focus on literature review of related work and defining the research gap. The authors should also streamline the manuscript’s Introduction section to make it more concise. In addition, the authors should carefully proofread their manuscript before the submission. Currently, the manuscript has many grammar issues and the reviewer suggests the authors review the manuscript with a professional editor.
2. The authors should provide more information about the equations that they provided in the manuscript. In the current format, the reviewer has been easily lost in comprehension. In addition, the authors should provide a Discussion section to discuss the research results. Moreover, the authors should proofread their equations to ensure that all symbols are explicitly explained.
3. The authors conducted a comprehensive maneuver simulation of the quadrotor to demonstrate the performance of the proposed formation strategies. The authors have provided the parameters of the quadrotor. However, the authors did not provide the reasons for the selection of these parameters. What is the rationale behind the selection?
4. The authors conducted the maneuver simulation of the quadrotor in the user-defined coordinate system. But what will the performance of the proposed formation strategies look like in a geographic coordinate system and projected coordinate system?
5. All figures need to be revised to make them more legible. For example, Figures 7 – 12 has to be revised to make the text more legible. The reviewer cannot read anything in the printed manuscript. The reviewer has to zoom in to 200% to read them. Revise all figures and please create vector images if necessary. In addition, please provide the legends for Figures 15 – 20. Although the authors provide a legend figure in Figure 13, it will help the readers understand the Figures 15 – 20 much easier.
Author Response
file attached

Reviewer 2 Report
I like this result in which the authors present a distributed consensus control of quadrotors based on xyz-space. The analysis of their approach is highly appropriate and supported by MALAB Simulink simulation.
I only have a few minor suggestions, say, page 1, in abstract: model predictive control(MPC), should have a space, i.e., control (MPC). Similar for the others. Also, line 22 on the same page: survey[1], again should have a space for it. Otherwise it is good to go.
Author Response
Thanks for the comments. Corrections are included in the revised manuscript. The revised manuscript is also edited by MDPI English editing service.
Round 2
Reviewer 1 Report
The authors have addressed my comments.